# PABI: A Unified PAC-Bayesian Informativeness Measure for Incidental Supervision Signals

## Abstract

Real-world applications often require making use of *a range of incidental supervision signals*. However, we currently lack a principled way to measure the benefit an incidental training dataset can bring, and the common practice of using indirect, weak signals is through exhaustive experiments with various models and hyperparameters. This paper studies whether we can, *in a single framework, quantify the benefit of various types of incidental signals for one's target task without going through combinatorial experiments*. We propose PABI, a unified informativeness measure motivated by PAC-Bayesian theory, characterizing the reduction in uncertainty that indirect, weak signals provide. We demonstrate PABI's use in quantifying various types of incidental signals including partial labels, noisy labels, constraints, cross-domain signals, and combinations of these. Experiments with various setups on two natural language processing (NLP) tasks, named entity recognition (NER) and question answering (QA), show that PABI correlates well with learning performance, providing a promising way to determine, ahead of learning, which supervision signals would be beneficial.

## 1 Introduction

The supervised learning paradigm, where direct supervision signals are assumed to be available in high-quality and large amounts, has been struggling to fulfill the needs in many real-world AI applications. As a result, researchers and practitioners often resort to datasets that are not collected directly for the target task but, hopefully, capture some phenomena useful for it (Pan & Yang, 2009; Vapnik & Vashist, 2009; Roth, 2017; Kolesnikov et al., 2019). However, it remains unclear how to predict the benefits of these incidental signals on our target task beforehand, so the common practice is often trial-and-error: do experiments with different combinations of datasets and learning protocols, often exhaustively, to achieve improvement on a target task (Liu et al., 2019; Khashabi et al., 2020). Not only this is very costly, this trial-and-error approach can also be hard to interpret: *if we don't see improvements, is it because the incidental signals themselves are not useful for our target task, or is it because the learning protocols we have tried are inappropriate?*

The difficulties of foreshadowing the benefits of various incidental supervision signals are two-fold. First, it is hard to provide a *unified* measure because of the intrinsic differences among different signals (e.g., how do we predict and compare the benefit of learning from noisy data and the benefit of knowing some constraints for the target task?). Second, it is hard to provide a *practical* measure supported by *theory*. Previous attempts are either not practical or too heuristic (Baxter, 1998; Ben-David et al., 2010; Thrun & O'Sullivan, 1998; Gururangan et al., 2020). In this paper, we propose a unified PAC-Bayesian motivated informativeness measure (PABI) to quantify the value of incidental signals. We suggest that the informativeness of various incidental signals can be uniformly characterized by the reduction in the original concept class uncertainty they provide. Specifically, in the PAC-Bayesian framework[1], the informativeness is based on the Kullback–Leibler (KL) divergence between the prior and the posterior, where incidental signals are used to estimate a better prior (closer to the gold posterior) to achieve better generalization performance. Furthermore, we provide a more practical entropy-based approximation of PABI. In practice, PABI first computes the entropy of the prior estimated from incidental signals, and then computes the relative decrease to the entropy of the prior without any information, as the informativeness of incidental signals.

---

[1]We choose the PAC-Bayes framework here because it allows us to link PABI to the performance measure.

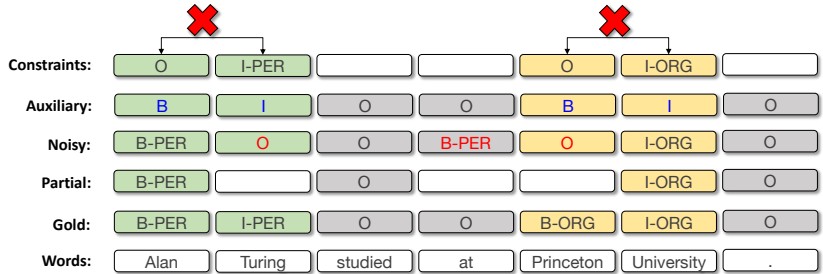

Figure 1: An example of NER with various incidental supervision signals: partial labels (some missing labels in structured outputs), noisy labels (some incorrect labels), auxiliary labels (labels of another task, e.g. named entity detection in the figure), and constraints in structured learning (e.g. the BIO constraint where I-X must follow B-X or I-X (Ramshaw & Marcus, 1999) in the figure).

We have been in need of a unified informativeness measure like `PABI`. For instance, it might be obvious that we can expect better learning performance if the training data are less noisy and more completely annotated, but what if we want to compare the benefits of a noisy dataset and that of a partial dataset? `PABI` enables this kind of comparisons beforehand, on a wide range of incidental signals such as partial labels, noisy labels, constraints[2], auxiliary signals, cross-domain signals, and some combinations of them, for sequence tagging tasks in NLP. A specific example of NER is shown in Fig. 1, and the advantages of `PABI` are in Table 1.

Finally, our experiments on two NLP tasks, NER and QA, show that there is a strong positive correlation between `PABI` and the relative improvement for various incidental signals. This strong positive correlation indicates that the proposed unified, theory-motivated measure `PABI` can serve as a good indicator of the final learning performance, providing a promising way to know which signals are helpful for a target task beforehand.

**Organization**. We start with related work in Section 1.1. Then we derive informativeness measure `PABI` in Section 2. We show examples on how to compute `PABI` using various incidental signals in Section 3. We verify the effectiveness of `PABI` in Section 4. Section 5 concludes this paper.

## 1.1 RELATED WORK

There are lots of practical measures proposed to quantify the benefits of specific types of signals. For example, a widely used measure for partial signals in structured learning is the partial rate (Cour et al., 2011; Hovy & Hovy, 2012; Liu & Dietterich, 2014; Van Rooyen & Williamson, 2017; Ning et al., 2019); a widely used measure for noisy signals is the noise ratio (Angluin & Laird, 1988; Natarajan et al., 2013; Rolnick et al., 2017; Van Rooyen & Williamson, 2017); Ning et al. (2019) propose to use the concaveness of the mutual information with different percentage of annotations to quantify the strength of constraints in the structured learning; others, in NLP, have quantified the contribution of constraints experimentally (Chang et al., 2012; 2008). Bjerva (2017) proposes to use conditional entropy or mutual information to quantify the value for auxiliary signals. As for domain adaptation, domain similarity can be measured by the performance gap between domains (Wang et al., 2019) or measures based on the language model in NLP, such as the vocabulary overlap (Gururangan et al., 2020). Among them, the most relevant work is (Bjerva, 2017). However, their conditional entropy or mutual information is based on token-level label distribution, which cannot be used for incidental signals involving multiple tokens or inputs, such as constraints and cross-domain signals. At the same time, for the cases where both `PABI` and mutual information can handle, `PABI` works similar to the mutual information as shown in Fig. 2, and `PABI` can further be shown to be a strictly increasing function of the mutual information. The key advantage of our proposed measure `PABI` is that `PABI` is a unified measure motivated by the PAC-Bayesian theory for a broader range of incidental signals compared to these practical measures for specific types of incidental signals.

There also has been a line of theoretical work that attempts to exploit incidental supervision signals. Among them, the most relevant part is task relatedness. Ben-David & Borbely (2008) define the

---

[2]Constraints are used to model the dependency among words and sentences, which are considered in a lot of work, such as CRF (Lafferty et al., 2001) and ILP (Roth & Yih, 2004).

| Measures | Incidental Supervision Signals | | | | | Unified Measure | | Support | |
|---|---|---|---|---|---|---|---|---|---|
| | partial | noisy | constraints | auxiliary | cross-domain | cross-type | mixed-type | Theoretical | Empirical |
| CST'11, HH'12, LD'14 | ✓ | × | × | × | × | × | × | ✓ | ✓ |
| AL'88, NDRT'13, RVBS'17 | × | ✓ | × | × | × | × | × | ✓ | ✓ |
| VW'17, WNR'20 | ✓ | ✓ | × | × | × | × | × | ✓ | × |
| NHFR'19 | ✓ | × | ✓ | × | × | × | × | × | ✓ |
| B'17 | × | × | × | ✓ | × | × | × | × | ✓ |
| GMSLBDS'20 | × | × | × | × | ✓ | × | × | × | ✓ |
| PABI (ours) | ✓ | ✓ | ✓ | ✓ | ✓ | ✓ | ✓ | ✓ | ✓ |

Table 1: Comparison between PABI and prior works: (1) Cross-type: PABI is a unified measure which can measure the benefit of different types of incidental signals (e.g., comparing a noisy dataset and a partially annotated dataset). (2) Mixed-type: PABI can measure the benefit of mixed incidental signals (e.g., a dataset that is both noisy and partially annotated). (3) PABI is derived from PAC-Bayesian theory but also easy to compute in practice; PABI is shown to have similar or better predicting capability of signals' benefit (see Figs. 2 and 3 and Sec. 3.1).[4]

task relatedness based on the richness of transformations between inputs for different tasks, but their analysis is limited to cases where data is from the same classification problem but the inputs are in different subspace. Juba (2006) proposes to use the joint Kolmogorov complexity (Li et al., 2008) to characterize relatedness, but it is still unclear how to compute the joint Kolmogorov complexity in real-world applications. Mahmud & Ray (2008) further propose to use conditional Kolmogorov complexity to measure the task relatedness and provide empirical analysis for decision trees, but it is unclear how to use their relatedness for other models, such as deep neural networks. Thrun & O'Sullivan (1998) propose to cluster tasks based on the similarity between the task-optimal distance metric of k-nearest neighbors (KNN), but their analysis is based on KNN and it is unclear how to use their relatedness for other models. A lot of other works provide quite good qualitative analysis to show various incidental signals are helpful but they did not provide quantitative analysis to quantify to what extent these types of incidental signals can help (Balcan & Blum, 2010; Abu-Mostafa, 1993; Natarajan et al., 2013; Van Rooyen & Williamson, 2017; Baxter, 1998; London et al., 2016; Ciliberto et al., 2019; Ben-David et al., 2010; Wang et al., 2020). Compared to these theoretical analyses, PABI can be easily used in practice to quantify the benefits of a broader range of incidental signals.

## 2 PABI: A UNIFIED PAC-BAYESIAN INFORMATIVENESS MEASURE

We start with notations and preliminaries. Let $\mathcal{X}$ be the input space, $\mathcal{Y}$ be the label space, and $\hat{\mathcal{Y}}$ be the prediction space. Let $\mathcal{D}$ denote the underlying distribution on $\mathcal{X} \times \mathcal{Y}$. Let $\ell : \mathcal{Y} \times \hat{\mathcal{Y}} \to \mathbb{R}_+$ be the loss function that we use to evaluate learning algorithms. A set of training samples $S = \{x_i, y_i\}_{i=1}^m$ is generated i.i.d. from $\mathcal{D}$. In the common supervised learning setting, we usually assume the concept that generates data comes from the concept class $\mathcal{C}$. In this paper, we assume $\mathcal{C}$ is finite, and its size is $|\mathcal{C}|$, which is common in NLP. We want to choose a predictor $c : \mathcal{X} \to \hat{\mathcal{Y}}$ from $\mathcal{C}$ such that it generalizes well to unseen data with respect to $\ell$, measured by the *generalization error* $R_{\mathcal{D}}(c) = \mathbb{E}_{\mathbf{x}, \mathbf{y} \sim \mathcal{D}}[\ell(\mathbf{y}, c(\mathbf{x}))]$. The *training error* over $S$ is $R_S(c) = \frac{1}{m} \sum_{i=1}^m \ell(y_i, c(x_i))$.

More generally, instead of predicting a concept, we can specify a distribution over the concept class. Let $\mathcal{P}$ denote the space of probability distributions over $\mathcal{C}$. General Bayesian learning algorithms (Zhang et al., 2006) in the PAC-Bayesian framework (McAllester, 1999a;b; Seeger, 2002; McAllester, 2003b;a; Maurer, 2004; Guedj, 2019) aim to choose a posterior $\pi_\lambda \in \mathcal{P}$ over the concept class $\mathcal{C}$ based on a prior $\pi_0 \in \mathcal{P}$ and training data $S$, where $\lambda$ is a hyper parameter that controls the tradeoff between the prior and the data likelihood. In this setting, the training error and the generalization error need to be generalized, to take the distribution into account, as $L_S(\pi_\lambda) = \mathbb{E}_{c \sim \pi_\lambda}[R_S(c)]$ and $L_{\mathcal{D}}(\pi_\lambda) = \mathbb{E}_{c \sim \pi_\lambda}[R_{\mathcal{D}}(c)]$ respectively. One can easily see that when the posterior is *one-hot* (exactly one entry of the distribution is 1), we have the original definitions of training error and generalization error, as in the PAC framework (Valiant, 1984).

---

[4]Papers are denoted by abbreviations of author names as: CST'11 (Cour et al., 2011), HH'12 (Hovy & Hovy, 2012), LD'14 (Liu & Dietterich, 2014), AL'88 (Angluin & Laird, 1988), NDRT'13 (Natarajan et al., 2013), RVBS'17 (Rolnick et al., 2017), VW'17 (Van Rooyen & Williamson, 2017), WNR'20 (Wang et al., 2020), NHFR'19 (Ning et al., 2019), B'17 (Bjerva, 2017), GMSLBDS'20 (Gururangan et al., 2020).

## 2.1 INFORMATIVENESS MEASURES IN THE PAC-BAYESIAN FRAMEWORK

We are now ready to derive the proposed informativeness measure PABI motivated by PAC-Bayes. The generalization error bound in the PAC-Bayesian framework (Guedj, 2019; Catoni, 2007) says that with probability $1 - \delta$ over $S$, $L_{\mathcal{D}}(\pi_{\lambda^*}) \leq L_{\mathcal{D}}(\pi^*) + \sqrt{\frac{8B(D_{KL}(\pi^*||\pi_0) + \ln \frac{2\ln(mC)}{\delta})}{m}}$, where $\pi_{\lambda^*}$ is the posterior distribution with the optimal $\lambda^* = \sqrt{\frac{2m(D_{KL}(\pi^*||\pi_0) + \ln \frac{2}{\delta})}{B}}$, $\pi^* \in \mathcal{P}$ is the gold posterior that generates the data, $D_{KL}(\pi^*||\pi_0)$ denotes the KL divergence from $\pi_0$ to $\pi^*$, $B$ and $C$ are two constants. This is based on the Theorem 2 in Guedj (2019).

As shown in the generalization bound, the generalization error is bounded by the KL divergence $D_{KL}(\pi^*||\pi_0)$ from the prior distribution to the gold posterior distribution. *Therefore, we propose to utilize incidental signals to improve the prior distribution from $\pi_0$ to $\tilde{\pi}_0$ so that it is closer to the gold posterior distribution $\pi^*$.* Correspondingly, we can define PABI, the informativeness measure for incidental supervision signals, by measuring the improvement with regard to the gold posterior, in KL divergence sense.

**Definition 2.1** (PABI). *Suppose we use incidental signals to improve the prior distribution from $\pi_0$ to $\tilde{\pi}_0$. The informativeness measure for incidental signals, PABI, is defined as*

$$S(\pi_0, \tilde{\pi}_0) \triangleq \sqrt{1 - \frac{D_{KL}(\pi^*||\tilde{\pi}_0)}{D_{KL}(\pi^*||\pi_0)}} \tag{1}$$

**Remark.** Note that $S(\pi_0, \tilde{\pi}_0) = 0$ if $\tilde{\pi}_0 = \pi_0$, while if $\tilde{\pi}_0 = \pi^*$, then $S(\pi_0, \tilde{\pi}_0) = 1$. This result is consistent with our intuition that the closer $\tilde{\pi}_0$ is to $\pi^*$, the more benefits we can gain from incidental signals. The square root function is used in PABI for two reasons: first, the generalization bounds in both PAC-Bayesian and PAC (see Sec. 2.2) frameworks have the square root function; second, in our later experiments, we find that square root function can significantly improve the Pearson correlation between the relative performance improvement and PABI. It is worthwhile to note that the square root is not crucial for our framework, because our goal is to compare the benefits among different incidental supervision signals, where the relative values are expressive enough. In this sense, any strictly increasing function in $[0, 1]$ over the current formulation would be acceptable.

In our paper, we focus on the setting that the gold posterior $\pi^*$ is *one-hot*, which means $\pi^*$ concentrates on the true concept $c^* \in \mathcal{C}$, though the definition of PABI can handle general gold posterior. However, $\pi^*$ is unknown in practice, which makes Eq. (1) hard to be computed in reality. In the following, we provide an approximation $\hat{S}$ of PABI.

**Definition 2.2** (Approximation of PABI). *Assume that the original prior $\pi_0$ is uniform, and the gold posterior $\pi^*$ is one-hot concentrated on the true concept $c^*$ in $\mathcal{C}$, as we have assumed that $\mathcal{C}$ is finite. Let $H(\cdot)$ be the entropy function. The approximation $\hat{S}$ of PABI is defined as*

$$\hat{S}(\pi_0, \tilde{\pi}_0) \triangleq \sqrt{1 - \frac{H(\tilde{\pi}_0)}{H(\pi_0)}} = \sqrt{1 - \frac{H(\tilde{\pi}_0)}{\ln|\mathcal{C}|}} \tag{2}$$

The uniform prior $\pi_0$ is usually used when we do not have information about the prior on which concept in the class that generates data. *The intuition behind $\hat{S}$ is that, it measures how much entropy incidental signals reduce, compared with non-informative prior $\pi_0$.* $\hat{S}$ can be computed through data and thus is practical. To see how this approximation works, first note $D_{KL}(\pi^*||\pi_0) = \ln|\mathcal{C}|$ because $\pi^*$ is one-hot and $\pi_0$ is uniform over the finite concept class $\mathcal{C}$. Let $\pi_c$ be the one-hot distribution concentrated on concept $c$ for each $c \in \mathcal{C}$. The approximation is that we estimate $\pi^*$ by $\pi_c$, where $c$ follows $\tilde{\pi}_0$: $D_{KL}(\pi^*||\tilde{\pi}_0) \approx \mathbb{E}_{c \sim \tilde{\pi}_0} D_{KL}(\pi_c||\tilde{\pi}_0)$. It turns out $\mathbb{E}_{c \sim \tilde{\pi}_0} D_{KL}(\pi_c||\tilde{\pi}_0) = H(\tilde{\pi}_0)$. Therefore,

$$\hat{S}(\pi_0, \tilde{\pi}_0) = \sqrt{1 - \frac{H(\tilde{\pi}_0)}{\ln|\mathcal{C}|}} = \sqrt{1 - \frac{\mathbb{E}_{c \sim \tilde{\pi}_0} D_{KL}(\pi_c||\tilde{\pi}_0)}{\ln|\mathcal{C}|}} \approx \sqrt{1 - \frac{D_{KL}(\pi^*||\tilde{\pi}_0)}{D_{KL}(\pi^*||\pi_0)}} = S(\pi_0, \tilde{\pi}_0).$$

We later show that the approximation of PABI and PABI is equivalent in the non-probabilistic cases with the finite concept class, indicating the quality of this approximation. Furthermore, the effectiveness of this approximation in NLP applications also indicates the quality of this approximation.

## 2.2 PABI IN THE PAC FRAMEWORK

We have derived PABI in the PAC-Bayesian framework. Here, we discuss briefly on what PABI reduces to in the PAC framework and what limitations are when PABI is restricted to the PAC framework. The generalization bound in the PAC framework (Mohri et al., 2018) says with probability $1 - \delta$ over $S$, $R_{\mathcal{D}}(c) \leq R_S(c) + \sqrt{\frac{\ln|\mathcal{C}| + \ln\frac{2}{\delta}}{2m}}$. *We propose to reduce the concept class from $\mathcal{C}$ to $\tilde{\mathcal{C}}$ by using incidental signals.* Then PABI in the PAC framework can be written as

$$S(\mathcal{C}, \tilde{\mathcal{C}}) = \sqrt{1 - \frac{\ln|\tilde{\mathcal{C}}|}{\ln|\mathcal{C}|}} \tag{3}$$

It turns out that Eq. (3) is a special case of Eq. (1) when $\pi^*$ is one-hot over $\mathcal{C}$, $\pi_0$ is uniform over $\mathcal{C}$ and $\tilde{\pi}_0$ is uniform over $\tilde{\mathcal{C}}$ (See Appx. A.1 for the derivation). As shown in Appx. A.1, we can see that the three informativeness measures, PABI in Eq. (1), the approximation of PABI in Eq. (2), and PABI in the PAC framework in Eq. (3), are equivalent, i.e. $S(\pi_0, \tilde{\pi}_0) = \hat{S}(\pi_0, \tilde{\pi}_0) = S(\mathcal{C}, \tilde{\mathcal{C}})$, in the non-probabilistic cases with the finite concept class. The equivalence among three measures further indicates that both PABI and the approximation of PABI are reasonable.

However, PABI restricted to the PAC framework cannot handle the probabilistic cases. For example, incidental signals can reduce the probability of some concepts, though the concept class is not reduced. In this example, $S(\mathcal{C}, \tilde{\mathcal{C}})$ is zero, but we actually benefit from incidental signals. Some analysis on the extensions and general limitations of PABI can be found in Appx. A.2 and A.3. We need to notice that the size of concept class also plays an important role in the lower bound on the generalization error (more details in Appx. A.4), indicating that PABI based on the reduction of the concept class is a reasonable measure.

## 3 EXAMPLES FOR PABI

In this section, we show some examples of sequence tagging tasks[5] in NLP for PABI. Similar to the categorization of transfer learning (Pan & Yang, 2009), we use *inductive signals* to denote the signals with a different conditional probability distribution ($P(\mathbf{y} \mid \mathbf{x})$) from gold signals, such as noisy and auxiliary signals, and *transductive signals* to denote the signals with the same task ($P(\mathbf{y} \mid \mathbf{x})$) as gold signals but a different marginal distribution of $\mathbf{x}$ ($P(\mathbf{x})$) from gold signals, such as cross-domain and cross-lingual signals. In our following analysis, we study the tasks with finite concept class which is quite common in NLP. For simplicity, we focus on simple cases where the number of incidental signals is large enough. How different factors (including base model performance, size of incidental signals, data distributions, algorithms, cost-sensitive losses) affect PABI are discussed in Appx. A.5. We derive the PABI for partial labels in detail and the derivations for others are similar. More examples and details can be found in Appx. A.6.

### 3.1 EXAMPLES WITH INDUCTIVE SIGNALS

**Partial labels.** The labels for each example in sequence tagging tasks is a sequence and some of them are unknown in this case. Assuming that the ratio of the unknown labels in data is $\eta_p \in [0, 1]$, the size of the reduced concept class will be $|\tilde{\mathcal{C}}| = |\mathcal{L}|^{n|\mathcal{V}|^n \eta_p}$. Therefore, $\hat{S}(\pi_0, \tilde{\pi}_0) = S(\pi_0, \tilde{\pi}_0) = S(\mathcal{C}, \tilde{\mathcal{C}}) = \sqrt{1 - \frac{\ln|\tilde{\mathcal{C}}|}{\ln|\mathcal{C}|}} = \sqrt{1 - \frac{|\mathcal{L}|^{n|\mathcal{V}|^n \eta_p}}{|\mathcal{L}|^{n|\mathcal{V}|^n}}} = \sqrt{1 - \eta_p}$. It is consistent with the widely used partial rate because it is a monotonically decreasing function of the partial rate.

**Noisy labels.** For each token, $P(y|\tilde{y})$ is determined by the noisy rate $\eta_n \in [0, 1]$, i.e. $P(y = \tilde{y}) = 1 - \eta_n$ and the probability of other labels are all $\frac{\eta_n}{|\mathcal{L}| - 1}$. We can get the corresponding probability distribution of labels over the tokens in all inputs ($\tilde{\pi}_0$ over the concept class). In this way, $\hat{S}(\pi_0, \tilde{\pi}_0) = \sqrt{1 - \frac{\eta_n \ln(|\mathcal{L}| - 1) - \eta_n \ln \eta_n - (1 - \eta_n) \ln(1 - \eta_n)}{\ln|\mathcal{L}|}}$. It is consistent with the widely used noise rate because it is a monotonically decreasing function of the noisy rate. In practice, the noisy

---

[5]Given an input $\mathbf{x} \in \mathcal{X} = \mathcal{V}^n$ generated from a distribution $\mathcal{D}$, the task aims to get the corresponding label $\mathbf{y} \in \hat{\mathcal{Y}} = \mathcal{Y} = \mathcal{L}^n$, where $\mathcal{V}$ is the vocabulary of input words, $\mathcal{L}$ is the label set for the task, and $n$ is the length of the input sentence.

rate can be easily estimated with some aligned data[6], and the noise with more complex patterns (e.g. input dependent) is postponed as our future work.

## 3.2 Examples with Transductive Signals

For transductive signals, such as cross-domain signals, we can first extend the concept class $\mathcal{C}$ to the extended concept class $\mathcal{C}^e$ with the corresponding extended input space $\mathcal{X}^e$. After that, we can use incidental signals to estimate a better prior distribution $\tilde{\pi}_0^e$ over the extended concept class $\mathcal{C}^e$, and then get the corresponding $\tilde{\pi}_0$ over the original concept class by restricting the concept from $\mathcal{X}^e$ to $\mathcal{X}$. In this way, the informativeness of transductive signals can still be measured by $S(\pi_0, \tilde{\pi}_0)$ or $\hat{S}(\pi_0, \tilde{\pi}_0)$. The restriction step is similar to Roth & Zelenko (2000).

However, how to compute $H(\tilde{\pi}_0)$ is still unclear. We now provide a way to estimate it. To better illustrate the estimation process for transductive signals, we provide the summary of core notations in Table 3 as shown in Appx. A.11. For simplicity, we use $c(\mathbf{x})$ to denote the gold system on the gold signals, $\tilde{c}(\mathbf{x})$ to denote the perfect system on the incidental signals, and $\hat{c}(\mathbf{x})$ to denote the silver system trained on the incidental signals. Source domain (target domain) is the domain of incidental signals (gold signals). We use $\mathcal{D}$ to denote the target domain and $\tilde{\mathcal{D}}$ to denote the source domain. $P_{\mathcal{D}}(\mathbf{x})$ is the marginal distribution of $\mathbf{x}$ under $\mathcal{D}$, and similar definition for $P_{\tilde{\mathcal{D}}}(\mathbf{x})$. In our analysis, we assume $\tilde{c}(\mathbf{x})$ is a noisy version of $c(\mathbf{x})$ with the noisy rate $\eta$, and $\hat{c}(\mathbf{x})$ is a noisy version of $\tilde{c}(\mathbf{x})$ with the noisy rate $\eta_1$ ($\eta_1'$) in the source (target) domain : $\eta_1 = \mathbb{E}_{\mathbf{x} \sim P_{\tilde{\mathcal{D}}}(\mathbf{x})} \mathbf{1}(\hat{c}(\mathbf{x}) \neq \tilde{c}(\mathbf{x}))$ ($\eta_1' = \mathbb{E}_{\mathbf{x} \sim P_{\mathcal{D}}(\mathbf{x})} \mathbf{1}(\hat{c}(\mathbf{x}) \neq \tilde{c}(\mathbf{x}))$).

In practice, $\eta$ is unknown but it can be estimated by $\eta_1$ in the source domain and $\eta_2 = \mathbb{E}_{\mathbf{x} \sim P_{\mathcal{D}}(\mathbf{x})} \mathbf{1}(\hat{c}(\mathbf{x}) \neq c(\mathbf{x}))$ (the noisy rate of the silver system compared to the gold system on the target domain) as follows:

$$\eta = \mathbb{E}_{\mathbf{x} \sim P_{\mathcal{D}}(\mathbf{x})} \mathbf{1}(c(\mathbf{x}) \neq \tilde{c}(\mathbf{x})) = \frac{(|\mathcal{L}| - 1)(\eta_1' - \eta_2)}{1 - |\mathcal{L}|(1 - \eta_1')} = \frac{(|\mathcal{L}| - 1)(\eta_1 - \eta_2)}{1 - |\mathcal{L}|(1 - \eta_1)}. \tag{4}$$

Here we add an assumption: $\eta_1'$ in the target domain is equal to $\eta_1$ in the source domain.[7] In Appx. A.7, we can see that Eq. (4) serves as an unbiased estimator for $\eta$ under some assumptions, but the concentration rate will depend on the size of source data. It requires finer-grained analysis on the estimator in Eq. (4), which we postpone as our future work. Similar to noisy labels, the corresponding informativeness of transductive signals can be then computed as $\hat{S}(\pi_0, \tilde{\pi}_0) = \sqrt{1 - \frac{\eta \ln(|\mathcal{L}| - 1) - \eta \ln \eta - (1 - \eta) \ln(1 - \eta)}{\ln |\mathcal{L}|}}$. Note that we treat cross-domain signals as a special type of noisy data, when $\eta$ is estimated.

To justify the use of $\eta$ in the informativeness measure for transductive signals, we show in Theorem A.2 (see Appx. A.8) that (informally speaking) the generalization error of a learner that is jointly trained on data from both source and target domains can be upper bounded by $\eta$ (plus a function of the size of the concept class and the number of samples). Finally we note that although the computation cost of PABI for transductive signals is higher than that for inductive signals, it is still much cheaper than building combined models with joint training. For example, given $T$ source domains and $T$ target domains, the goal is to select the best source domain for each target domain. If we use the joint training, we need to train $T \times T = T^2$ models. However, with PABI, we only need to train $T + T = 2T$ models. Furthermore, for each model, joint training on the combination of two domains requires more time than the training on a single domain used in PABI. In this situation, we can see that PABI is much cheaper than building combined models with joint training.

## 3.3 Examples with Mixed Incidental Signals

**The mix of partial and noisy labels.** The corresponding informativeness for the mix of partial and noisy labels is $\hat{S}(\pi_0, \tilde{\pi}_0) = \sqrt{(1 - \eta_p) * (1 - \frac{\eta_n \ln(|\mathcal{L}| - 1) - \eta_n \ln \eta_n - (1 - \eta_n) \ln(1 - \eta_n)}{\ln |\mathcal{L}|})}$, where $\eta_p \in [0, 1]$ denotes the ratio of unlabeled tokens, and $\eta_n \in [0, 1]$ denotes the noise ratio.

---

[6] If there is no jointly annotated data, we can use similar methods as Bjerva (2017) to create some approximate jointly annotated data.

[7] We add this assumption mainly because we want to estimate the $\eta$ only based on $\eta_1$ and $\eta_2$, which can be easily computed in practice.

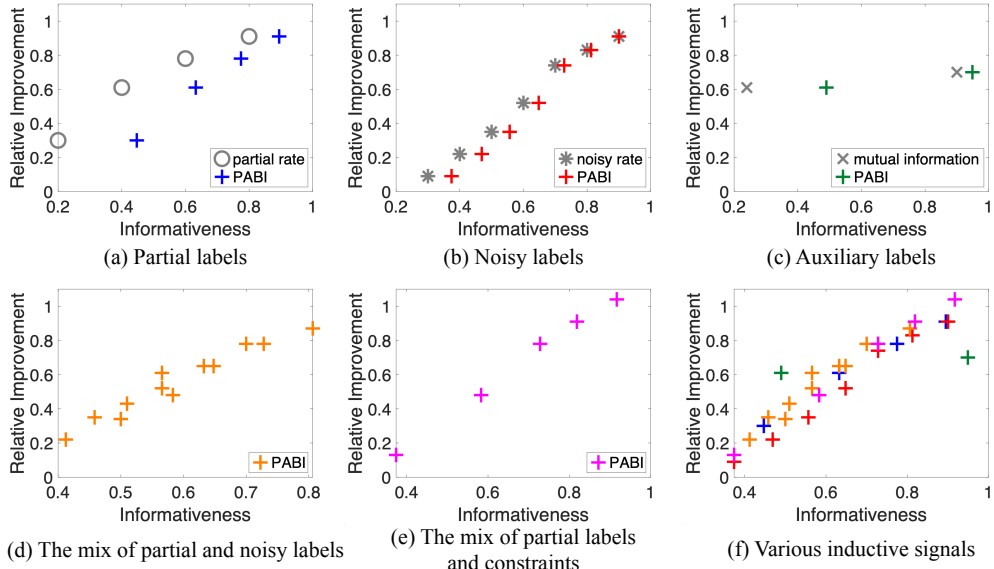

Figure 2: Correlations between informativeness and relative performance improvement for NER with various inductive signals. **On one hand, as shown in (a)-(c), `PABI` has similar foreshadowing ability with measures for specific signals. On the other hand, as shown in (d)-(f), `PABI` can measure the benefits of mixed inductive signals and compare different types of inductive signals, which cannot be handled by existing frameworks.** For individual inductive signals, the baselines (gray points) are, i.e. one minus partial rate for partial labels (Cour et al., 2011; Hovy & Hovy, 2012; Liu & Dietterich, 2014; Van Rooyen & Williamson, 2017; Ning et al., 2019), one minus noisy rate for noisy labels (Angluin & Laird, 1988; Natarajan et al., 2013; Rolnick et al., 2017; Van Rooyen & Williamson, 2017), and entropy normalized mutual information for auxiliary labels (Bjerva, 2017). For NER with various inductive signals (f) (with all `PABI` points from (a)-(e)), Pearson's correlation and Spearman's rank correlation are 0.92 and 0.93. Note that the relative improvement for NER (with informativeness 0.90 but relative improvement 0.70) in auxiliary labels (c) is smaller than expected mainly due to the **imbalanced label distribution** (88% O among all BIO labels). More discussions about the imbalanced distribution can be found in Appx. A.5.

**The mix of partial labels and constraints.** For BIO constraints with partial labels, we can use dynamic programming with sampling as Ning et al. (2019) to estimate $\ln |\tilde{\mathcal{C}}|$ and $\hat{S}(\pi_0, \tilde{\pi}_0)$.

## 4 EXPERIMENTS

In this section, we verify the effectiveness of `PABI` on various inductive signals and transductive signals on NER and QA. More details about experimental settings are in Appx. A.9.

**Learning with various inductive signals.** In this part, we analyze the informativeness of inductive signals for NER. We use Ontonotes NER (18 types of named entities) (Hovy et al., 2006) as the main task. We randomly sample 10% sentences (30716 words) of the development set as the small gold signals, 90% sentences (273985 words) of the development set as the large incidental signals. We use a two-layer NNs with 5-gram features as our basic model. The lower bound for our experiments is the result of the model with small gold Ontonotes NER annotations and bootstrapped on the unlabeled texts of the large gold Ontonotes NER, which is 38 F1, and the upper bound is the result of the model with both small gold Ontonotes NER annotations and the large gold Ontonotes NER annotations, which is 61 F1. To utilize inductive signals, we propose a new bootstrapping based algorithm CWBPP (Algorithm 1 in Appx. A.10), where inductive signals are used to improve the inference stage by approximating a better prior. It is an extension of CoDL (Chang et al., 2007) with various inductive signals.

We experiment on NER with various inductive signals, including three types of individual signals, partial labels, noisy labels, auxiliary labels, and two types of mixed signals: signals with both partial

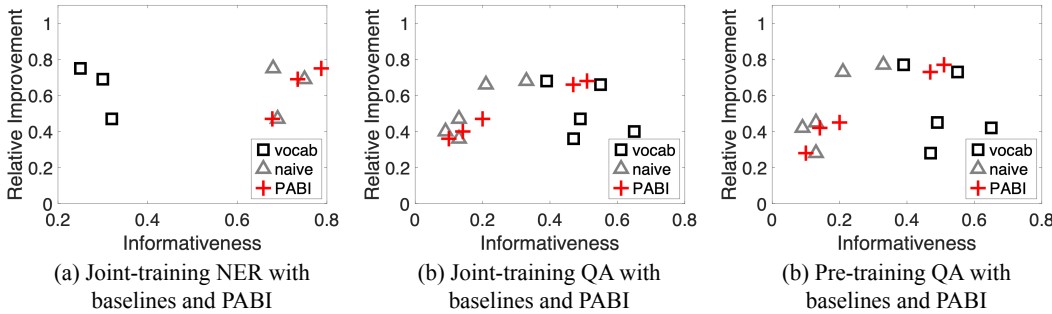

(a) Joint-training NER with baselines and PABI

(b) Joint-training QA with baselines and PABI

(b) Pre-training QA with baselines and PABI

Figure 3: Correlation between informativeness measures (baselines or the PABI) and relative performance improvement (via joint training or pre-training) for cross-domain NER and cross-domain QA. **We can see that the correlation between the relative improvement and PABI is stronger than other baselines.** Red results with the PABI which is based on $\eta$ in Eq. (4); Gray points indicate the results with the naive informativeness measure $\eta_2$; Black points indicate the results with the vocabulary overlap baseline (Gururangan et al., 2020). The Pearson's correlation of three informativeness measure in the three cases are: $0.96/0.19/-0.85$, $1.00/0.88/-0.40$, $0.99/0.85/-0.30$, indicating the quality of the PABI measure. Similarly, the corresponding Spearman's rank correlation are: $1.00/-0.50/-1.00$, $1.00/0.82/-0.30$, $1.00/0.82/-0.30$.

and noisy labels, and signals with both partial labels and constraints. As shown in Fig. 2, we find that there is a strong correlation between the relative improvement and PABI for various inductive signals. For individual signals in Fig. 2(a)-2(c), we find that PABI have similar foreshadowing ability comparing to the measures for specific signals, i.e., $1 - \eta_p$ for partial labels, $1 - \eta_n$ for noisy labels, and entropy normalized mutual information[8] ($\frac{I(Y;\tilde{Y})}{H(Y)}$) for auxiliary labels. For mixed signals in Fig. 2(d)-2(e), the strong correlation is quite promising because the benefits of mixed signals cannot be quantified by existing frameworks. Finally, the strong positive correlation for different types of signals in Fig. 2(f) indicates that it is feasible to compare the benefits of different incidental signals with PABI, which cannot be addressed by existing frameworks.

**Learning with cross-domain signals.** In this part, we consider the benefits of cross-domain signals for NER and QA. For NER, we consider four NER datasets, Ontonotes, CoNLL, twitter (Strauss et al., 2016), and GMB (Bos et al., 2017). We aim to detect the person names here because the only shared type of the four datasets is the person[9]. In our experiments, the twitter NER serves as the main dataset and other three datasets are cross-domain datasets. There are 85 sentences in the small gold training set, 756 sentences (9 times of the gold signals) in the large incidental training set, and 851 sentences in the test set. We tried larger datasets for both gold and incidental signals (keeping the ratio between two sizes as 9) and the results are similar as long as the number of gold signals is not too large. For QA, we consider SQuAD (Rajpurkar et al., 2016), QAMR (Michael et al., 2017), QA-SRL Bank 2.0 (FitzGerald et al., 2018), QA-RE (Levy et al., 2017), NewsQA (Trischler et al., 2017), TriviaQA (Joshi et al., 2017). In our experiments, the SQuAD dataset serves as the main dataset and other datasets are cross-domain datasets. We randomly sample 700 QA pairs as the small gold signals, about $6.2K$ QA pairs as the large incidental signals (9 times of the small gold signals), and $21K$ QA pairs as the test data.

We use BERT as our basic model and consider two strategies to make use of incidental signals: joint training and pre-training. For NER, the lower bound is the result with only small gold twitter annotations, which is 61.51 F1, and the upper bound is the result with both small gold twitter annotations and large gold twitter annotations, which is 78.31. For QA, the lower bound is the result with only small gold SQuAD annotations, which is 26.45 exact match. The upper bound for the joint training is the result with both small gold SQuAD annotations and large SQuAD annotations, which is 50.72 exact match. Similarily, the upper bound for the pre-training is 49.24 exact match.

---

[8] In Bjerva (2017), they propose to use mutual information or conditional entropy to measure the informativeness, so we normalize the mutual information with the entropy to make the value between 0 and 1.

[9] Note that our focus here is cross-domain signals, the divergent set of classes for different domains is a mix of cross-domain and auxiliary signals, which is our future work.

The relation between the relative improvement (pre-training or joint training) and informativeness measures (baselines or the `PABI`) is shown in Fig. 3. We can see that there is a strong positive correlation between the relative improvement and `PABI` for cross-domain signals. Comparing to the naive baseline $\eta_2$, we can see that the adjustment from $\eta_1$ is crucial (Eq. (4)), indicating that directly using $\eta_2$ is not a good choice. We also show the vocabulary overlap baseline as in (Gururangan et al., 2020) where we compute the overlap over the top $1K$ most frequent unigrams (excluding stop words and punctuations) between different domains. The results for this baseline are quite bad, and the fact that our data is not so large makes this baseline more valueless.

## 5   CONCLUSION AND FUTURE WORK

Motivated by PAC-Bayesian theory, this paper proposes a unified framework, `PABI`, to characterize incidental supervision signals by how much uncertainty they can reduce in the hypothesis space. We demonstrate the effectiveness of `PABI` in foreshadowing the benefits of various signals, i.e., partial labels, noisy labels, auxiliary labels, constraints, cross-domain signals and combinations of them, for solving NER and QA. To our best knowledge, `PABI` is the first informativeness measure that can handle various incidental signals and combinations of them; `PABI` is motivated by PAC-Bayes and can be easily computed in real-world tasks. As the recent success of natural language modeling has given rise to many explorations in knowledge transfer across tasks and corpora (Bjerva, 2017; Phang et al., 2018; Zhu et al., 2019; Liu et al., 2019; He et al., 2020; Khashabi et al., 2020) , `PABI` is a concrete step towards explaining some of these observations.

We conclude our work by pointing out several interesting directions for our future work.

First, `PABI` can also provide guidance in designing learning protocols. For instance, in a B/I/O sequence chunking task,[10] missing labels make it a partial annotation problem, while treating missing labels as O introduces noise. Since the informativeness of partial signals is larger than that of noisy signals with the same partial/noisy rate (see details in Sec. 3.1), `PABI` suggests us *not* to treat missing labels as O, and this is exactly what Mayhew et al. (2019) prove to us via their experiments. We plan to explore more in this direction to apply `PABI` in designing better learning protocols.

Second, we need to acknowledge that our current exploration for auxiliary labels is still limited. The results for auxiliary labels with a different label set (Fig. 2(c)) is blocked by the imbalanced label distribution (Appx. A.5). For more complex cases, such as part-of-speech tagging (PoS) for NER, we can only treat them as cross-sentence constraints now and the results are also limited (Appx. A.6). In future, we will work more in this direction to better quantify the value of auxiliary signals.

Another interesting direction is to link `PABI` with the generalization bound. It might be too hard to directly link `PABI` with the generalization bound for all types of incidental signals, but it is possible to link it to the generalization bound for some specific types. For example, for partial and noisy labels, `PABI` can directly be expressed in the generalization bound as in (Cour et al., 2011; Natarajan et al., 2013; Van Rooyen & Williamson, 2017; Wang et al., 2020). The main difficulties are in constraints and auxiliary signals, and we postpone it as our future work.

Finally, we plan to evaluate `PABI` in more applications, such as textual entailment and image classification, and more types of signals, such as cross-lingual and cross-modal signals.

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

# A APPENDIX

## A.1 PABI IN THE PAC FRAMEWORK

Suppose $\pi^*$ is one-hot over $\mathcal{C}$, $\pi_0$ is uniform over $\mathcal{C}$ and $\tilde{\pi}_0$ is uniform over $\tilde{\mathcal{C}}$. We have $D_{KL}(\pi^*||\pi_0) = \ln|\mathcal{C}|$ and $D_{KL}(\pi^*||\tilde{\pi}_0) = \ln|\tilde{\mathcal{C}}|$. It follows that

$$S(\pi_0, \tilde{\pi}_0) = \sqrt{1 - \frac{D_{KL}(\pi^*||\tilde{\pi}_0)}{D_{KL}(\pi^*||\pi_0)}} = \sqrt{1 - \frac{\ln|\tilde{\mathcal{C}}|}{\ln|\mathcal{C}|}} = S(\mathcal{C}, \tilde{\mathcal{C}}).$$

At the same time, we have

$$\hat{S}(\pi_0, \tilde{\pi}_0) = \sqrt{1 - \frac{H(\tilde{\pi}_0)}{H(\pi_0)}} = \sqrt{1 - \frac{H(\tilde{\pi}_0)}{\ln|\mathcal{C}|}} = \sqrt{1 - \frac{\ln|\tilde{\mathcal{C}}|}{\ln|\mathcal{C}|}} = S(\mathcal{C}, \tilde{\mathcal{C}}).$$

Therefore, in the non-probabilistic cases with the finite concept class, the three informativeness measures are equivalent as:

$$S(\pi_0, \tilde{\pi}_0) = \hat{S}(\pi_0, \tilde{\pi}_0) = S(\mathcal{C}, \tilde{\mathcal{C}}).$$

The equivalence among three measures further indicates that both PABI and the approximation of PABI are reasonable.

## A.2 INFORMATIVENESS MEASURES IN PARAMETRIC CONCEPT CLASS

In practice, algorithms are often based on parametric concept class. The two informativeness measures in the PAC-Bayesian framework, $S(\pi_0, \tilde{\pi}_0)$ and $\hat{S}(\pi_0, \tilde{\pi}_0)$, can be easily adapted to handle the cases in parametric concept class. Given parametric space $\mathcal{C}_w$, we can easily change the probability distribution $\pi(\mathcal{C}_w)$ over the parametric concept class to the probability distribution $\pi(\mathcal{C})$ over the finite concept class $\mathcal{C} = \{c : \mathcal{V}^n \to \mathcal{L}^n\}$ by clustering concepts in the parametric space according to their outputs on all inputs. The concepts in each cluster have the same outputs on all inputs as outputs of one concept in the finite concept class $\mathcal{C}$. We then merge the probabilities of concepts in the same cluster to get the probability distribution $\pi(\mathcal{C})$ over the finite concept class $\mathcal{C}$. This merging approach can be applied to any concept class which is not equal to the finite concept class $\mathcal{C}$, including non-parametric and semi-parametric concept class. In practice, we can use sampling algorithms, such as Markov chain Monte Carlo (MCMC) methods, to simulate this clustering strategy.

## A.3 LIMITATIONS OF INFORMATIVENESS MEASURES

Different informativeness measures are based on different assumptions, so we analyze their limitations in detail to understand their limitations in applications.

For the informativeness measure $S(\mathcal{C}, \tilde{\mathcal{C}})$, it cannot handle probabilistic signals or infinite concept classes. There are various probabilistic incidental signals, such as soft constraints and probabilistic co-occurrences between an auxiliary task and the main task. An example of probabilistic co-occurrences between part-of-speech (PoS) tagging and NER is that the adjectives have a $95\%$ probability to have the label $O$ in NER. As for the infinite concept class, most classifiers are based on infinite parametric spaces. Thus, $S(\mathcal{C}, \tilde{\mathcal{C}})$ cannot be applied to these classifiers.

The informativeness measure $S(\pi_0, \tilde{\pi}_0)$ is hard to be computed for some complex cases. In practice, we can use the estimated posterior distribution over the gold data, which is asymptotically unbiased, to estimate it. Another approximation is to use the informativeness measure $\hat{S} = \sqrt{1 - \frac{H(\tilde{\pi}_0)}{H(\pi_0)}}$. However, it is not directly linked to the generalization bound, so more work is needed to guarantee its reliability for some complex probabilistic cases. We postpone to provide the theoretical guarantees for $\hat{S} = \sqrt{1 - \frac{H(\tilde{\pi}_0)}{H(\pi_0)}}$ on more complex cases as our future work.

## A.4 LOWER BOUND IN THE PAC FRAMEWORK

In the following theorem, we show that the VC dimension (size of concept class) also plays an important role in the lower bund for the generalization error, indicating that PABI based on the reduction of the concept class is a reasonable measure.

**Theorem A.1.** *Let $\mathcal{C}$ be a concept class with VC dimension $d > 1$. Then, for any $m \geq 1$ and any learning algorithm $\mathcal{A}$, there exists a distribution $\mathcal{D}$ over $\mathcal{X}$ and a target concept $c \in \mathcal{C}$ such that*

$$P_{S \sim \mathcal{D}^m}[R_{\mathcal{D}}(c_S) > \frac{d-1}{32m}] \geq 1/100$$

*where $c_S$ is a consistent concept with S returned by $\mathcal{A}$. This is the Theorem 3.20 in Chapter 3.4 of Mohri et al. (2018).*

### A.5 DISCUSSION OF SOME OTHER FACTORS IN PABI

In this subsection, we consider the impact of the following factors in PABI: base model performance, the size of incidental signals, data distribution, algorithm and cost-sensitive loss.

**Base model performance.** In the generalization bound in both PAC and PAC-Bayesian, we can see that the relative improvement in the generalization bound from reducing $\mathcal{C}$ is small if $m$ is large. In practice, the relative improvement is the real improvement with some noise. Therefore, we can see that the real improvement is dominant if $m$ is small and the noise is dominant if $m$ is large. Therefore, PABI may not work well when $m$ is large and when the performance on the target task is already good enough.

**The size of incidental signals.** Our previous analysis is based on a strong assumption that incidental signals are large enough (ideally $\tilde{m} \to \infty$). A more realistic PABI is based on $\tilde{\mathcal{C}}$ with $\tilde{m}$ examples as $S(\mathcal{C}, \tilde{\mathcal{C}}) = \sqrt{\frac{\ln |\mathcal{C}_{\tilde{m}}| - \ln |\tilde{\mathcal{C}}_{\tilde{m}}|}{\ln |\mathcal{C}|}} = \sqrt{\frac{\ln |\mathcal{C}_{\tilde{m}}| - \ln |\tilde{\mathcal{C}}_{\tilde{m}}|}{\ln |\mathcal{C}_{\tilde{m}}|} \times \frac{\ln |\mathcal{C}_{\tilde{m}}|}{\ln |\mathcal{C}|}} = \sqrt{(1 - \frac{\ln |\tilde{\mathcal{C}}_{\tilde{m}}|}{\ln |\mathcal{C}_{\tilde{m}}|}) \times \frac{\ln |\mathcal{C}_{\tilde{m}}|}{\ln |\mathcal{C}|}} = \sqrt{(1 - \frac{\ln |\tilde{\mathcal{C}}|}{\ln |\mathcal{C}|}) \times \frac{\ln |\mathcal{C}_{\tilde{m}}|}{\ln |\mathcal{C}|}}$, where $\mathcal{C}_{\tilde{m}}$ denotes the restricted concept class of $\mathcal{C}$ on the $\tilde{m}$ examples, and so does $\tilde{\mathcal{C}}_{\tilde{m}}$. Note that the ratio of the intrinsic information in incidental signals is independent of the size $\tilde{m}$, so $\frac{\ln |\tilde{\mathcal{C}}_{\tilde{m}}|}{\ln |\mathcal{C}_{\tilde{m}}|} = \frac{\ln |\tilde{\mathcal{C}}|}{\ln |\mathcal{C}|}$ holds for our signals. For example, $\frac{\ln |\tilde{\mathcal{C}}_{\tilde{m}}|}{\ln |\mathcal{C}_{\tilde{m}}|} = \eta_p$ for partial data with unknown ratio $\eta_p$, doesn't depend on the size $\tilde{m}$. (1) When $\tilde{m}$ is large enough, $S(\mathcal{C}, \tilde{\mathcal{C}}) = \sqrt{1 - \frac{\ln |\tilde{\mathcal{C}}|}{\ln |\mathcal{C}|}}$. (2) When the sizes of different incidental signals are all $\tilde{m}$, the relative improvement is independent of $\tilde{m}$ because $\frac{\ln |\mathcal{C}_{\tilde{m}}|}{\ln |\mathcal{C}|}$ is the same constant for different incidental signals. Our experiments are based on this case and does not really rely on the assumption that incidental signals are large enough. (3) The incidental signals we are comparing are not large enough and have different sizes, we need to use $S(\mathcal{C}, \tilde{\mathcal{C}}) = \sqrt{(1 - \frac{\ln |\tilde{\mathcal{C}}|}{\ln |\mathcal{C}|}) \times \frac{\tilde{m}}{|\mathcal{V}|^n}}$ to incorporate that difference. We can replace $|\mathcal{V}|^n$ with some reasonable $M$, e.g. the largest size of incidental signals, to make PABI in a larger range of values in [0, 1]. In future, we need to explore more in this direction.

**Data distribution.** As for the distribution of examples, both PAC and PAC-Bayesian are distribution-free (see more in Chapter 2.1 of Mohri et al. (2018)). However, if we consider the joint distribution between examples and labels, such as imbalanced label distribution, the situation will be different. Specific types of joint data distribution refer to a restricted concept class $\mathcal{C}'$. Therefore, PABI is expected to work well if the reduction from $\mathcal{C}$ is similar to the reduction from $\mathcal{C}'$ with incidental signals, i.e. $S(\mathcal{C}', \tilde{\mathcal{C}}') = \sqrt{1 - \frac{\ln |\tilde{\mathcal{C}}'|}{\ln |\mathcal{C}'|}} \approx \sqrt{1 - \frac{\ln |\tilde{\mathcal{C}}|}{\ln |\mathcal{C}|}}$.

**Algorithm.** Different algorithms make different assumptions on the concept class. For example, SVM aims to find the maximum-margin hyperplane (see more in Chapter 5.4 of Mohri et al. (2018)). Therefore, a specific algorithm actually is based on a restricted concept class $\mathcal{C}'$ (e.g. concepts with margin in SVM case). Similarly, PABI is expected to work well if the reduction from $\mathcal{C}$ is similar to the reduction from $\mathcal{C}'$ with incidental signals. We also cannot compare the benefits from various incidental signals with different algorithms. If the algorithm is not expressive enough to take advantage of incidental signals, we may also not be able to use PABI there.

**Cost-sensitive Loss.** For different loss functions other than 0-1 loss, there are still some similar generalization bounds in PAC and PAC-Bayesian (using complexity of concept class and sample size) (Bartlett et al., 2006; Ciliberto et al., 2016). Therefore, PABI can also be used (possibly with some minor modifications) for cost-sensitive loss functions.

| k-gram | 1 | 2 | 3 | 4 | 5 | 6 | 7 | 8 | 9 | 10 |
|---|---|---|---|---|---|---|---|---|---|---|
| word-pos | 8.68 | 49.45 | 84.08 | 96.22 | 98.96 | 99.54 | 99.69 | 99.73 | 99.75 | 99.76 |
| word-ner | 27.65 | 76.23 | 92.98 | 98.04 | 99.37 | 99.74 | 99.84 | 99.88 | 99.89 | 99.90 |
| pos-ner | 0.20 | 6.65 | 13.78 | 25.36 | 41.50 | 60.14 | 77.04 | 88.61 | 95.01 | 97.92 |
| ner-pos | 0.00 | 0.01 | 0.03 | 0.07 | 0.17 | 0.39 | 0.80 | 1.47 | 2.45 | 3.71 |

Table 2: K-gram co-occurrence analysis for PoS and NER in the whole Ontonotes dataset. For example, word-pos represents the percentage of k-gram words that have the unique k-gram PoS labels.

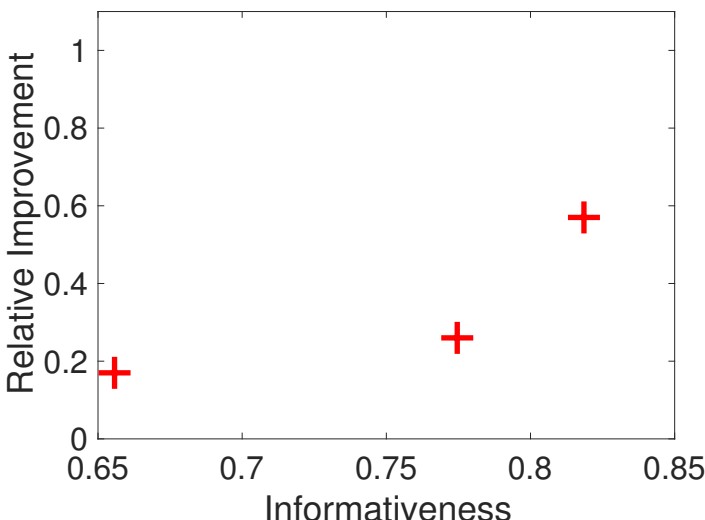

Figure 4: The correlations between the informativeness and the relative performance improvement for NER with cross-sentence constraints.

### A.6 MORE EXAMPLES WITH INCIDENTAL SIGNALS

In this subsection, we show more examples with incidental signals, including within-sentence constraints, cross-sentence constraints, auxiliary labels, cross-lingual signals, cross-modal signals, and the mix of cross-domian signals and constraints.

**Within-Sentence Constraints.** As for within-sentence constraints, we show three types of common constraints in NLP, which are BIO constraints, assignment constraints, and ranking constraints.

- BIO constraints are widely used in sequence tagging tasks, such as NER. For BIO constraints, I-X must follow B-X or I-X, where "X" is finer types such as PER (person) and LOC (location). We consider a simple case here: there are only B, I, O three labels. We have $\ln|\tilde{\mathcal{C}}| = |\mathcal{V}|^n(\ln|\mathcal{L}|^n + \ln[\sum_{m=0}^{\lfloor(n+1)/2\rfloor}\binom{m}{n-m+1}(\frac{-1}{|\mathcal{L}|^2})^m])$ for the BIO constraint. Therefore, $\hat{S}(\pi_0, \tilde{\pi}_0) = S(\pi_0, \tilde{\pi}_0) = S(\mathcal{C}, \tilde{\mathcal{C}}) = \sqrt{1 - \frac{\ln|\mathcal{L}|^n + \ln[\sum_{m=0}^{\lfloor(n+1)/2\rfloor}\binom{m}{n-m+1}(\frac{-1}{|\mathcal{L}|^2})^m]}{\ln|\mathcal{L}|^n}}$. This value can be approximated by the dynamic programming as Ning et al. (2019).

- Assignment constraints can be used in various types of semantic parsing tasks, such as semantic role labeling (SRL). Assume we need to assign $d$ agents with $d'$ tasks such that the agent nodes and the task nodes form a bipartite graph (without loss of generality, assume $d \le d'$). Each agent is represented by a feature vector in $\mathcal{V}_f$. We have $\hat{S}(\pi_0, \tilde{\pi}_0) = S(\pi_0, \tilde{\pi}_0) = S(\mathcal{C}, \tilde{\mathcal{C}}) = \sqrt{1 - \frac{\ln|\tilde{\mathcal{C}}|}{\ln|\mathcal{C}|}} = \sqrt{1 - \frac{\ln\binom{d}{d'}}{d\ln d'}}$. This informativeness doesn't rely on the choice of $\mathcal{V}_f$ where that $\mathcal{V}_f$ denotes discrete feature space for arguments.

- Ranking constraints can be used in ranking problems, such as temporal relation extraction. For a ranking problem with $t$ items, there are $d = t(t-1)/2$ pairwise comparisons in total. Its structure is a chain following the transitivity constraints, i.e., if $A < B$ and $B < C$, then $A < C$. In this way, we have $\hat{S}(\pi_0, \tilde{\pi}_0) = S(\pi_0, \tilde{\pi}_0) = S(\mathcal{C}, \tilde{\mathcal{C}}) = \sqrt{1 - \frac{\ln |\tilde{\mathcal{C}}|}{\ln |\mathcal{C}|}} = \sqrt{1 - \frac{\ln t!}{\ln 2^d}} \approx \sqrt{1 - \frac{2 \ln t - 2}{(t-1) \ln 2}}$. This informativeness doesn't rely on the choice of $\mathcal{V}_f$ where $\mathcal{V}_f$ denotes discrete feature space for events.

**Cross-sentence Constraints.** For cross-sentence constraints, we consider a common example, global statistics based on 2-tuple of tokens, i.e. pairs of tokens in different sentences must have the same labels. We can group words into $K$ groups with probability $p$. In this way, we have $\hat{S}(\pi_0, \tilde{\pi}_0) = \sqrt{1 - \frac{-p \ln p - (1-p) \ln(1-p) + p \ln |\mathcal{L}|^K + (1-p) \ln(|\mathcal{L}|^n |\mathcal{V}|^n - |\mathcal{L}|^K))}{\ln |\mathcal{C}|}} \approx \sqrt{p}$. The approximation holds as long as $|\mathcal{L}|$, $\mathcal{V}$, and n are not all too small. For example, as shown in Table 2, the percentage of 5-gram words with unique NER labels is 99.37, so ideally the corresponding `PABI` will be $\sqrt{0.9937} = 0.9968$. It is worthwhile to note that the k-gram words with unique labels can also be caused by the low frequency of the appearance of the k-grams. In our experiments, we only consider the k-grams with unique labels that appear at least twice in the data. We experiment on NER with three types of cross-sentence constraints: uni-gram words with unique NER labels, bi-gram words with unique NER labels, and 5-gram part-of-speech (PoS) tags with unique NER labels[11]. The results are shown in Fig. 4.

**Auxiliary labels.** For auxiliary labels, we show two examples as follows:

- For a multi-class sequence tagging task, we use the corresponding detection task as auxiliary signals. Given a multi-class sequence tagging task with $C$ labels in the BIO format (Ramshaw & Marcus, 1999), we will have 3 labels for the detection and $2C + 1$ labels for the classification. Thus, $\hat{S}(\pi_0, \tilde{\pi}_0) = S(\pi_0, \tilde{\pi}_0) = S(\mathcal{C}, \tilde{\mathcal{C}}) = \sqrt{1 - \frac{(1-p_o) \ln C}{\ln(2C+1)}}$, where $p_o$ is the percentage of the label O among all labels.

- Coarse-grained NER for Fine-grained NER. We have four types, PER, ORG, LOC and MISC for CoNLL NER and 18 types for Ontonotes NER. The mapping between CoNLL NER and Ontonotes NER is as follows: PER (PERSON), ORG (ORG), LOC(LOC, FAC, GPE), MISC(NORP, PRODUCT, EVENT, LANGUAGE), O(WORF_OF_ART, LAW, DATE, TIME, PERCENT, MONEY, QUANTITY, ORDINAL, CARDINAL, O) (Augenstein et al., 2017). In the BIO setting, we have $\hat{S}(\pi_0, \tilde{\pi}_0) = S(\pi_0, \tilde{\pi}_0) = S(\mathcal{C}, \tilde{\mathcal{C}}) = \sqrt{1 - \frac{P_l \ln 3 + P_m \ln 4 + P_o \ln 19}{\ln 37}}$, where $p_l$, $p_m$, $p_o$ are the percentage of LOC(including B-LOC and I-LOC), MISC (including B-MISC and I-MISC), and O among all possible labels.

Note that `PABI` is consistent with the entropy normalized mutual information (see more in footnote 8) because $\hat{S}(\pi_0, \tilde{\pi}_0) = \sqrt{\frac{I(Y; \tilde{Y})}{H(Y)}}$ for auxiliary labels.

**Cross-lingual signals.** For cross-lingual signals, we can use multilingual BERT to get $\hat{c}$ in the extended input space $(\mathcal{V} \cup \mathcal{V}')^n$. After that, $\eta_1$ and $\eta_2$ can be computed accordingly.

**Cross-modal signals.** For cross-modal signals, we only consider the case where labels of gold and incidental signals are same and inputs of gold and incidental are aligned. A common situation is that a video has visual, acoustic, and textual information. In this case, the images and speech related to the texts can be used as cross-modal information. We can use cross-modal mapping between speech/images and texts (e.g. Chung et al. (2018)) to estimate the $\eta_1$ and $\eta_2$ for cross-modal signals.

**The mix of cross-domain signals and constraints.** Let $\tilde{c}$ denote the perfect system on cross-domain signals and satisfying constrains on inputs of gold signals, and $\hat{c}$ denote the model trained on cross-domain signals and satisfying constraints on inputs of gold signals. In this way, we can estimate $\eta_1$ and $\eta_2$ by forcing constraints in their inference stage.

---

[11]Here we use PoS tags as a special type of cross-sentence constraints by specifying the labels of tokens whose PoS tags have unique NER labels, although PoS tags can also be viewed as auxiliary signals for NER.

## A.7 DERIVATION OF EQUATION (4)

For simplicity, we use $Y$ to denote $c(\mathbf{x})$, $\tilde{Y}$ to denote $\tilde{c}(\mathbf{x})$, and $\hat{Y}$ to denote $\hat{Y}(\mathbf{x})$. We then re-write the definitions of $\eta$, $\eta_1'$ and $\eta_2$ as $\eta = \mathbb{E}_{\mathbf{x} \sim P_{\mathcal{D}}(\mathbf{x})} \mathbf{1}(c(\mathbf{x}) \neq \tilde{c}(\mathbf{x})) = P(Y \neq \tilde{Y})$, $\eta_1' = \mathbb{E}_{\mathbf{x} \sim P_{\mathcal{D}}(\mathbf{x})} \mathbf{1}(\hat{c}(\mathbf{x}) \neq \tilde{c}(\mathbf{x})) = P(\hat{Y} \neq \tilde{Y})$ and $\eta_2 = \mathbb{E}_{\mathbf{x} \sim P_{\mathcal{D}}(\mathbf{x})} \mathbf{1}(\hat{c}(\mathbf{x}) \neq c(\mathbf{x})) = P(\hat{Y} \neq Y)$. Note that $\mathcal{L}$ is the label set for the task. Considering all three systems in the target domain, we have

$$
\begin{aligned}
1 - \eta_2 &= P(\hat{Y} = Y) \\
&= P(\hat{Y} = Y, \tilde{Y} = Y) + P(\hat{Y} = Y, \tilde{Y} \neq Y) \\
&= P(\tilde{Y} = Y)P(\hat{Y} = Y | \tilde{Y} = Y) + P(\tilde{Y} \neq Y)P(\hat{Y} = Y | \tilde{Y} \neq Y) \\
&= P(\tilde{Y} = Y)P(\hat{Y} = \tilde{Y}) + P(\tilde{Y} \neq Y)\frac{P(\hat{Y} \neq \tilde{Y})}{|\mathcal{L}| - 1} \\
&= (1 - \eta)(1 - \eta_1') + \frac{\eta \eta_1'}{|\mathcal{L}| - 1}
\end{aligned}
$$

Therefore, we have $\eta = \frac{(|\mathcal{L}| - 1)(\eta_1' - \eta_2)}{1 - |\mathcal{L}|(1 - \eta_1')}$.

## A.8 PABI FOR TRANSDUCTIVE SIGNALS

Assumption I: $\tilde{c}(\mathbf{x})$ is a noisy version of $c(\mathbf{x})$ with a noise ratio $\eta$ in both target and source domain: $\eta = \mathbb{E}_{\mathbf{x} \sim P_{\mathcal{D}}(\mathbf{x})} \mathbf{1}(c(\mathbf{x}) \neq \tilde{c}(\mathbf{x})) = \mathbb{E}_{\mathbf{x} \sim P_{\tilde{\mathcal{D}}}(\mathbf{x})} \mathbf{1}(c(\mathbf{x}) \neq \tilde{c}(\mathbf{x}))$.

**Theorem A.2.** *Let $\mathcal{C}$ be a concept class of VC dimension $d$ for binary classification. Let $S_+$ be a labeled sample of size $m$ generated by drawing $\beta m$ points (S) from $\mathcal{D}$ according to $c$ and $(1 - \beta)m$ points ($\tilde{S}$) from $\tilde{D}$ (the distribution of incidental signals) according to $\tilde{c}$. If $\hat{c}' = \arg \min_{c \in \mathcal{C}} R_{S_+, \frac{1}{2}}(c) = \arg \min_{c \in \mathcal{C}} \frac{1}{2} R_S(c) + \frac{1}{2} R_{\tilde{S}}(c)$ is the empirical joint error minimizer, and $c_T^* = \arg \min_{c \in \mathcal{C}} R_{\mathcal{D}}(c)$ is the target error minimizer, $c^* = \arg \min_{c \in \mathcal{C}} R_{\tilde{\mathcal{D}}}(c) + R_{\mathcal{D}}(c)$ is the joint error minimizer, under assumption I, and assume that $\mathcal{C}$ is expressive enough so that both the target error minimizer and the joint error minimizer can achieve zero errors, then for any $\delta \in (0, 1)$, with probability at least $1 - \delta$,*

$$
R_{\mathcal{D}}(\hat{c}') \leq \eta + 4\sqrt{\frac{1}{\beta} + \frac{1}{1 - \beta}} \sqrt{\frac{2d \ln \frac{2em}{d} + 2 \ln \frac{8}{\delta}}{m}}
$$

A concept is a function $c \colon \mathcal{X} \to \{0, 1\}$. The probability according to the distribution $\mathcal{D}$ that a concept $c$ disagrees with a labeling function $f$ (which can also be a concept) is defined as

$$
R_{\mathcal{D}}(c, f) = \mathbb{E}_{\mathbf{x} \in \mathcal{D}}[|c(\mathbf{x}) - f(\mathbf{x})|] \tag{5}
$$

Note that here $\ell(y, c(x)) = |y - c(x)|$ is the loss function and $R_{\mathcal{D}}(c) = \mathbb{E}_{\mathbf{x} \sim \mathcal{D}}[\ell(y, c(\mathbf{x}))]$ where $y$ is the gold label for $\mathbf{x}$. We denote $R_\alpha(c)$ ($\alpha \in [0, 1]$) the corresponding weighted combination of true source and target errors, measured with respect to $\tilde{\mathcal{D}}$ and $\mathcal{D}$ as follows:

$$
R_\alpha(c) = \alpha R_{\mathcal{D}}(c) + (1 - \alpha) R_{\tilde{\mathcal{D}}}(c)
$$

**Lemma A.3.** *Let $c$ be a concept in concept class $\mathcal{C}$. Then*

$$
|R_\alpha(c) - R_{\mathcal{D}}(c)| \leq (1 - \alpha)(\Lambda + \tau(c))
$$

*where $\Lambda = R_{\tilde{\mathcal{D}}}(c^*) + R_{\mathcal{D}}(c^*)$, $c^* = \arg \min_{c \in \mathcal{C}} R_{\tilde{\mathcal{D}}}(c) + R_{\mathcal{D}}(c)$, and $\tau(c) = |R_{\tilde{\mathcal{D}}}(c, c^*) - R_{\mathcal{D}}(c, c^*)|$.*

*Proof.*

$$
\begin{aligned}
|R_\alpha(c) - R_{\mathcal{D}}(c)| &= (1 - \alpha)|R_{\tilde{\mathcal{D}}}(c) - R_{\mathcal{D}}(c)| \\
&\leq (1 - \alpha)[|R_{\tilde{\mathcal{D}}}(c) - R_{\tilde{\mathcal{D}}}(c, c^*)| + |R_{\tilde{\mathcal{D}}}(c, c^*) - R_{\mathcal{D}}(c, c^*)| + |R_{\mathcal{D}}(c, c^*) - R_{\mathcal{D}}(c)|] \\
&\leq (1 - \alpha)[R_{\tilde{\mathcal{D}}}(c^*) + |R_{\tilde{\mathcal{D}}}(c, c^*) - R_{\mathcal{D}}(c, c^*)| + R_{\mathcal{D}}(c^*)] \\
&= (1 - \alpha)(\Lambda + \tau(c))
\end{aligned}
$$

**Lemma A.4.** *For a fixed concept $c$ from $\mathcal{C}$ with VC dimension $d$, if a random labeled sample $(S_+)$ of size $m$ is generated by drawing $\beta m$ points $(S)$ from $\mathcal{D}$ and $(1-\beta)m$ points $(\tilde{S})$ from $\tilde{\mathcal{D}}$, and labeling them according to $f_{\mathcal{D}}$ and $f_{\tilde{\mathcal{D}}}$ respectively, then for any $\delta \in (0,1)$ with probability at least $1-\delta$ (over the choice of the samples),*

$$|R_\alpha(c) - R_{S_+,\alpha}(c)| \le 2\sqrt{\frac{\alpha^2}{\beta} + \frac{(1-\alpha)^2}{1-\beta}}\sqrt{\frac{2d\ln\frac{2em}{d} + 2\ln\frac{4}{\delta}}{m}}$$

*where $R_{S_+,\alpha} = \alpha R_S(c) + (1-\alpha)R_{\tilde{S}}(c)$ and $e$ is the natural number.*

*Proof.* Given Lemma 5 in Ben-David et al. (2010), which says for any $\delta \in (0,1)$, with probability $1-\delta$ (over the choice of the samples),

$$P[|R_{S_+,\alpha}(c) - R_\alpha(c)| \ge \epsilon] \le 2\exp\left(\frac{-2m\epsilon^2}{\frac{\alpha^2}{\beta} + \frac{(1-\alpha)^2}{1-\beta}}\right)$$

According to the Vapnik-Chervonenkis theory (Vapnik & Chervonenkis, 2015), we have with probability $1-\delta$,

$$|R_\alpha(c) - R_{S_+,\alpha}(c)| \le 2\sqrt{\frac{\alpha^2}{\beta} + \frac{(1-\alpha)^2}{1-\beta}}\sqrt{\frac{2d\ln\frac{2em}{d} + 2\ln\frac{4}{\delta}}{m}}$$

This is the standard generalization bound with an adjust term $\sqrt{\frac{\alpha^2}{\beta} + \frac{(1-\alpha)^2}{1-\beta}}$ (see more in Chapter 3.3 of Mohri et al. (2018)). $\qquad\square$

*Proof of Theorem A.2.* Let $\alpha = \frac{1}{2}$, then $\hat{c}' = \arg\min R_{S_+,\alpha}(c) = \frac{1}{2}(R_{\tilde{S}}(c) + R_S(c))$

$R_{\mathcal{D}}(\hat{c}') \le R_\alpha(\hat{c}') + (1-\alpha)(\Lambda + \tau(\hat{c}'))$ (Lemma A.3)

$\qquad \le R_\alpha(\hat{c}') + (1-\alpha)(\Lambda + |R_{\tilde{\mathcal{D}}}(\hat{c}', c^*) - R_{\mathcal{D}}(\hat{c}', c^*)|)$ (Definition of $\tau(\hat{c}')$)

$\qquad \le R_\alpha(\hat{c}') + (1-\alpha)(\Lambda + R_{\tilde{\mathcal{D}}}(\hat{c}') + R_{\tilde{\mathcal{D}}}(c^*) + R_{\mathcal{D}}(\hat{c}') + R_{\mathcal{D}}(c^*))$

$\qquad \le R_\alpha(\hat{c}') + (1-\alpha)(2\Lambda + 2R_\alpha(\hat{c}'))$

$\qquad = (3 - 2\alpha)R_\alpha(\hat{c}') + 2(1-\alpha)\Lambda$

$\qquad \le (3 - 2\alpha)(R_{S_+,\alpha}(\hat{c}') + 2\sqrt{\frac{\alpha^2}{\beta} + \frac{(1-\alpha)^2}{1-\beta}}\sqrt{\frac{2d\ln\frac{2em}{d} + 2\ln\frac{8}{\delta}}{m}})$

$\qquad + 2(1-\alpha)\Lambda$ (Lemma A.4 with $\delta/2$)

$\qquad \le (3 - 2\alpha)(R_{S_+,\alpha}(c_T^*) + 2\sqrt{\frac{\alpha^2}{\beta} + \frac{(1-\alpha)^2}{1-\beta}}\sqrt{\frac{2d\ln\frac{2em}{d} + 2\ln\frac{8}{\delta}}{m}})$

$\qquad + 2(1-\alpha)\Lambda$ ($\hat{c}' = \arg\min R_{S_+,\alpha}(c)$)

$\qquad \le (3 - 2\alpha)(R_\alpha(c_T^*) + 4\sqrt{\frac{\alpha^2}{\beta} + \frac{(1-\alpha)^2}{1-\beta}}\sqrt{\frac{2d\ln\frac{2em}{d} + 2\ln\frac{8}{\delta}}{m}})$

$\qquad + 2(1-\alpha)\Lambda$ (Lemma A.4 with $\delta/2$)

$\qquad \le (3 - 2\alpha)(R_{\mathcal{D}}(c_T^*) + 4\sqrt{\frac{\alpha^2}{\beta} + \frac{(1-\alpha)^2}{1-\beta}}\sqrt{\frac{2d\ln\frac{2em}{d} + 2\ln\frac{8}{\delta}}{m}} + (1-\alpha)(\Lambda + \tau(c_T^*)))$

$\qquad + 2(1-\alpha)\Lambda$ (Lemma A.3)

$\qquad \le (3 - 2\alpha)(R_{\mathcal{D}}(c_T^*) + 4\sqrt{\frac{\alpha^2}{\beta} + \frac{(1-\alpha)^2}{1-\beta}}\sqrt{\frac{2d\ln\frac{2em}{d} + 2\ln\frac{8}{\delta}}{m}})$

$\qquad + (2\alpha^2 - 7\alpha + 5)\Lambda + (2\alpha^2 - 5\alpha + 3)\tau(c_T^*)$

Note that

$\tau(c_T^*) = |R_{\tilde{\mathcal{D}}}(c_T^*, c^*) - R_{\mathcal{D}}(c_T^*, c^*)| \le R_{\tilde{\mathcal{D}}}(c_T^*) + R_{\tilde{\mathcal{D}}}(c^*) + R_{\mathcal{D}}(c_T^*) + R_{\mathcal{D}}(c^*) = \Lambda + R_{\mathcal{D}}(c_T^*) + R_{\tilde{\mathcal{D}}}(c_T^*)$

Therefore,

$$R_{\mathcal{D}}(\hat{c}') \leq R_{\tilde{\mathcal{D}}}(c_T^*) + 4\sqrt{\frac{1}{\beta} + \frac{1}{1-\beta}}\sqrt{\frac{2d\ln\frac{2em}{d} + 2\ln\frac{8}{\delta}}{m}} + 3R_{\mathcal{D}}(c_T^*) + 3\Lambda \quad (\alpha = \frac{1}{2})$$

Also note that $L_1$ loss is equivalent to 0-1 loss in the binary classification, so that $R_{\tilde{\mathcal{D}}}(c_T^*) = \eta$ under assumption I. In addition, assuming that $\mathcal{C}$ is expressive enough so that both the target error minimizer and the joint error minimizer can achieve zero errors ($R_{\mathcal{D}}(c_T^*) = 0$ and $\Lambda = 0$), the generalization bound can be simplified as follows:

$$R_{\mathcal{D}}(\hat{c}') \leq \eta + 4\sqrt{\frac{1}{\beta} + \frac{1}{1-\beta}}\sqrt{\frac{2d\ln\frac{2em}{d} + 2\ln\frac{8}{\delta}}{m}} \quad \square$$

Note that the proof of Theorem A.2 is similar to Theorem 3 in Ben-David et al. (2010). Our theorem is based on binary classification mainly because the error item in Eq. (5) based on the L1 loss will be equivalent to zero-one loss for binary classification. Although for multi-class classification, the L1 loss is different from commonly used zero-one loss, Theorem A.2 also indicates the relation between the generalization bound of joint training and the cross-domain performance $R_{\mathcal{D}}(c_S^*)$ (equal to $R_{\tilde{\mathcal{D}}}(c_T^*)$ under assumption I). Furthermore, a multi-class classification task can be represented by a series of binary classification tasks. Therefore, we postpone more accurate analysis for multi-class classification as our future work.

## A.9 DETAILS OF EXPERIMENTAL SETTINGS

In this subsection, we briefly highlight some important settings in our experiments and more details can be found in our submitted code.

**NER with individual inductive signals.** For partial labels, we experiment on NER with four different partial rates: 0.2, 0.4, 0.6, and 0.8. For noisy labels, we experiment on NER with seven different noisy rates: $0.1 - 0.7$. For auxiliary labels, we experiment on two auxiliary tasks: named entity detection and coarse NER (CoNLL annotations with 4 types of named entities (Sang & De Meulder, 2003)).

**NER with mixed inductive signals.** A more complex case is the comparison between the mixed inductive signals. For the first type of mixed signals, we experiment on the combination between three unknown partial rates (0.2, 0.4, and 0.6) and four noisy rates (0.1, 0.2, 0.3, and 0.4). As for the second type of mixed signals, we experiment on the combination between the BIO constraint and five unknown partial rates (0.2, 0.4, 0.6, 0.8, and 1.0).

**NER with various inductive signals.** After we put the three types of individual inductive signals and the two types of mixed inductive signals together, we still see a correlation between `PABI` and the relative performance improvement in experiments in Fig. 2(f).

**NER with cross-domain signals** Because we only focus on the person names, a lot of sentences in the original dataset will not include any entities. We random sample sentences to keep that 50% sentences without entities and 50% sentences with at least one entity. $\eta_1$ and $\eta_2$ is computed by using sentence-level accuracy.

**QA with cross-domain signals.** For consistency, we only keep one answer for each question in all datasets. Another thing worthwhile to notice is that the most informative QA dataset is not always the same for different main QA datasets. For example, for NewsQA, the most informative QA dataset is SQuAD, while the most informative QA dataset for SQuAD is QAMR.

**Experimental settings for learning with various inductive signals.** The 2-layer NNs we use in CWBPP (algorithm 1) has a hidden size of 4096, ReLU non-linear activation and cross-entropy loss. As for the embeddings, we use 300 dimensional Glove embeddings (Pennington et al., 2014). The size of the training batch is 10000 and the optimizer is Adam (Kingma & Ba, 2015) with learning rate $3e^{-4}$. When we initialize the classifier with gold signals (line 1), the number of training epochs is 20. After that, we conduct the bootstrapping 5 iterations (line 3-7). The confidence for predicted labels is exactly the predicted probability of the classifier (line 5). In each iteration of bootstrapping, we further train the classifier on the joint data 1 epoch (line 7).

---

**Algorithm 1:** Confidence-Weighted Bootstrapping with Prior Probability. The algorithm utilizes incidental signals to improve the inference stage in semi-supervised learning.

---

**Input:** A small dataset with gold signals $\mathcal{D} = (X_1, Y_1)$, and a large dataset with inductive signals $\tilde{\mathcal{D}} = (X_2, \tilde{Y}_2)$ where $X_1 \cap X_2 = \phi$

1 Initialize claissifier $\hat{c} = \text{LEARN}(\mathcal{D})$ (initialize the classifier with gold signals)
2 $P(Y_2|X_2, \tilde{Y}_2) = \text{PRIOR}(\mathcal{D}, \tilde{\mathcal{D}})$ (estimate the probability of gold labels for inputs in $\tilde{D}$)
3 **while** *convergence criteria not satisfied* **do**
4     $\hat{Y} = \text{INFERENCE}(X_2; \hat{c}; P(Y_2|X_2, \tilde{Y}_2))$ (get predicted labels of inputs in $\tilde{D}$)
5     $\hat{\rho} = \text{CONFIDENCE}(X_2; \hat{c}, P(Y_2|X_2, \tilde{Y}_2))$ (get confidence for predicted labels)
6     $\tilde{D} = (X_2, \hat{Y}, \hat{\rho})$ (get confidence-weighted incidental dataset with predicted labels)
7     $\hat{c} = \text{LEARN}(\mathcal{D} + \tilde{\mathcal{D}})$ (learn a classifier with both gold dataset and incidental dataset)
8 **return** $\hat{c}$

---

| Notations | Descriptions |
|---|---|
| $\mathcal{D}$ | target domain with gold signals |
| $\tilde{\mathcal{D}}$ | source domain with incidental signals |
| $c(\mathbf{x})$ | gold system on gold signals |
| $\tilde{c}(\mathbf{x})$ | perfect system on incidental signals |
| $\hat{c}(\mathbf{x})$ | silver system trained on incidental signals |
| $\eta$ | difference between the perfect system and the gold system in the target domain |
| $\eta_1$ | difference between the silver system and the perfect system in the source domain |
| $\eta_1'$ | difference between the silver system and the perfect system in the target domain |
| $\eta_2$ | difference between the silver system and the gold system in the target domain |

Table 3: Summary of core notations in the estimation process for transductive signals.

**Experimental settings for learning with cross-domain signals.** As for BERT, we use the pre-trained case-insensitive BERT-base pytorch implementation (Wolf et al., 2019). We use the common parameter settings for our experiments. Specifically, for NER, the max length is 256, batch size is 8, the epoch number is 4 and the learning rate is $5e^{-5}$. As for QA, the max length is 384, bath size is 16, the epoch number is 4, and the learning rate is $5e^{-5}$.

### A.10 THE CWBPP ALGORITHM

The CWBPP algorithm is shown in Algorithm 1.

### A.11 SUMMARY OF NOTATIONS IN THE ESTIMATION PROCESS FOR TRANSDUCTIVE SIGNALS

The summary of core notations in the estimation process for transductive signals is in Table 3.

