# OpenReview forum: "PABI: A Unified PAC-Bayesian Informativeness Measure for Incidental Supervision Signals"
_ICLR.cc/2021/Conference — Reject_

### Official Review · AnonReviewer1 · 2020-10-26
**A nice unified framework to measure informativeness for various incidental supervision signals, but with the lack of qualitative evaluation of the correlation studies.**

**Rating:** 5
**Confidence:** 3

**Review:**

##########################################################################
Summary:
This paper proposes a unified PAC-Bayesian-based informativeness measure (PABI) to quantify the value of incidental signals. PABI can measure various types of incidental signals such as partial labels, noisy labels, constraints, auxiliary signals, cross-domain signals, and their combinations. In NER and QA tasks, they showed the strong correlation signals between PABI and the relative improvements for various incidental signals.
##########################################################################
Reasons for score:

Overall, my score is marginally below than acceptance threshold.
Pros:
1. I enjoyed reading the paper, and I like the idea of covering various types of supervision signals at one unified measure.
2. The definition and approximation of PABI and its generalization to different inductive signals look sound to me.

Cons:

1. My biggest concern about this paper is the lack of clarity and presentation. In the introduction, I do understand how conceptually PABI is different from others, but do not know what it is. It would be better describing how PABI works in the introduction, Also, it would be better understanding the Section 2 and 3, if authors provide high-level insights of why each part of PABI’s description is important. Similarly, in the experiment, it was quite difficult to follow the text and capture the main claim. For instance, it would be better to understand if how Figure 2 and 3 should look like first and what trends of the points support the main claim of PABI, etc. Similarly, visual interpretation without specific guidelines make Figure 3 really difficult to understand. I guess some quantitative numbers would be very helpful like the linear regression slope, etc.

2.  Besides the presentation, I don’t quite understand how PABI can be used as a practical measure for other applications. Does the strong correlation with relative improvement mean that it can be used as an alternative measure of mutual information and further applied to other applications using such information measures in their optimization? If so, it would be nice to describe potential applications of these measures and other benefits of PABI in general. This also requires additional experiments that show its effectiveness in other applications.

#########################################################################
Some typos:
a widely used measure for for noisy signals -> a widely used measure for noisy signals
the SQuAD dataset servers as the main dataset -> the SQuAD dataset serves as the main dataset

---

> ### Author Response · Authors · 2020-11-13
> **Thanks for your insightful comments!**
>
> **Reply to “the lack of clarity and presentation”:** Thanks for your suggestions! **As for the presentation, we modified the introduction and figure captions as you suggested in the revision.**  Please let us know if you think the revision is not enough. **As for the quantitative numbers for Fig. 3, we have actually already provided Pearson's correlation** (measuring the linear correlation between two variables) and **Spearman’s rank correlation** (measuring the statistical dependence between the rankings of two variables) as shown in the caption of Fig. 3.
>
> **Reply to “the effectiveness in other applications”:** Thanks for your suggestions! **As for the effectiveness of PABI in other applications, we want to highlight that PABI (Definition 2.1) is quite general and can be applied to most real-world applications.** We evaluate PABI on NER because NER is a standard task in sequence tagging. Given the effectiveness of PABI in NER, it is likely that PABI can be used in most sequence tagging tasks, such as part-of-speech tagging, syntactic chunking, supersense tagging, and so on. QA is crucial for natural language understanding and has a lot of datasets. We believe that the effectiveness of PABI in QA can have some impact on natural language understanding. As for other tasks, such as textual entailment and image classification, we postpone evaluating PABI on them as our future work. **As for the alternative measure of mutual information, mutual information is actually a special case of PABI.**  In other words, PABI can be applied to a broader range of incidental signals compared to the mutual information (also other existing measures in the related work). For example, the mutual information in Bjerva (2017) is based on token-level label distribution, which cannot be used for incidental signals involving multiple tokens or inputs, such as constraints and cross-domain signals. At the same time, for the cases where both PABI and mutual information can handle, PABI works similar to the mutual information as shown in Figure 2, where PABI can further be shown to be a strictly increasing function of the mutual information.
>
> **We modified our paper in the revision, based on your suggestions. Please let us know if you think the revision is not enough.**

---

### Official Review · AnonReviewer4 · 2020-10-28
**Great work, minor comments on framing/pitch**

**Rating:** 8
**Confidence:** 3

**Review:**

This paper proposes PABI (PAC-Bayesian Informativeness?), a way of measuring and predicting the usefulness of “incidental supervision signal” for a downstream classification task. In particular, when labeled data is only available in noisy or partial form, or over a different domain than the target test domain, this data may still be used to improve a classifier, but it’s unclear how to tell which forms of incidental supervision will be most useful. Having a measure which allows us to compare different types of such supervision enables us to make intelligent tradeoffs.

PABI is proposed as a very general framework. The most general form of the measure, dealing with updates to the concept class prior, seems that it could capture any kind of incidental supervision. However, this means most of the work is in understanding how to apply and approximate it. This paper provides several such methods, particularly focusing on “inductive” learning (from constraints or partial/noisy gold labels) or “transductive” learning (from complete gold labels on different input domains). Mathematical developments of PABI are given for these cases, and experiments show that PABI is nicely positively correlated with the relative improvement that comes with various methods for integrating incidental supervision signal (including one which is developed as a side note by the authors).

Computing PABI may be challenging in some cases. In the case of transductive learning, it seems that a model needs to be trained on the incidental signal, although this is better than the combinatorial explosion of jointly trained models that would be required to test relative improvements directly. However, it’s not clear if efficient approximations for PABI will be feasible in all cases. This and other questions about the breadth of application of PABI are left for future work.

### Strengths

I think this paper is very well-motivated, situates itself well with respect to previous work, and presents clear advantages. Having a unified framework for comparing the utility of different kinds of incidental supervision signals seems potentially very useful, especially these days when incidental supervision of various sorts is instrumental in state-of-the-art models. It is also extremely relevant for data annotation and task design, which often has to make tradeoffs between these factors (i.e., noise versus partial annotation or dataset size).

There is a lot of content in this paper, including mathematical developments, algorithms, and experimental results. While I did not carefully check the proofs in the appendix, and I am not familiar with PAC-Bayesian theory or the associated literature, the paper seems technically sound to me.

### Weaknesses

While the generality of the proposed PABI framework is great and improves over existing work, I think this paper could be scoped more carefully and the scope could be clarified better.

As proposed, the PABI framework seems very general—which is good. But the paper only shows how to realize the framework in a couple specific cases, for “inductive” and “transductive” learning independently. This is still more general than previous work, but from the first few pages of the paper I was expecting something even more general.

* It seems to me that the combination of inductive and transductive learning may be possible using something close to the paper's proposed methods , but this isn’t addressed by the paper except a glancing mention in Footnote 6.
* It also is not clear to me from the paper’s text whether something close to the PABI framework can apply in broader settings like language modeling style pretraining, where the input-output format of the incidental supervision signal is different than that of the target task. In particular, it seems that in this case the approximation method proposed for transductive learning would indeed have to reduce to training a combined model. Related issues were finally mentioned briefly in the last paragraph of the paper, and something along these lines appears in appendix A.3, but I think a more up-front clarification of the limitations is warranted.

More broadly, the question in the back of my head when I began reading the paper was if this would help explain why and when language model pretraining (and other more flexible related-task pretraining) works well. The paper points to related work in this area, such as Gururangan et al 2020 (“Don’t Stop Pretraining”), leading me to think this paper would shed light on the issue, but in the end the issue was not mentioned and seems perhaps out of scope.

This is fine. All I would ask of the authors is to be more explicit about the limitations of PABI (or the proposed realizations of it) from the beginning, laying out the scope of this work and stating the limitations outright instead of only pointing to the appendix. It seems to me like PABI is more of a foundational framework which is ideal for future work to build into, rather than already being a general solution in itself. I think it would be best to pitch the paper this way.

### Recommendation

Accept. Important problem, lots of solid content, clear benefits over previous work and directions for the future. Great work.

### More comments/questions

I think the point of the formulation in Section 2.2 can be made a bit more explicit. It seems like the point is for applying PABI to partial labels. If that’s true (or there’s more to it) then might as well just say it there, or at least give this case as a motivating example.

Regarding the cross-domain results: why are the incidental supervision sets so small? It seems that there is a ton more incidental supervision available for NER, and in both cases the incidental supervision data is even smaller than the test set. Why not use more? It seems to me that the use case here is when a large amount of incidental supervision is available anyway. It also seems like the low-data setting is not totally fair to the vocabulary overlap baseline.

### Typos, style, etc.

When describing your experiments, I think it’s worth mentioning that they are on English text.

Figure 3: I don’t understand which numbers correspond to which model in the caption. This would be much easier to read in a table.

* P. 7: something’s wrong with “twitter(Strauss et al., 2016)”
* P. 7: The FitzGerald et al 2018 dataset is called “QA-SRL Bank 2.0”.
* P. 7: servers -> serves
* P. 7: “the lower bound for is”

---

> ### Author Response · Authors · 2020-11-13
> **Thanks for your insightful comments! In particular, we really appreciate your suggestions on the scope of the paper.**
>
> **Reply to “the scope of the paper”:** Thanks a lot for your kind suggestions. We will present the paper in a clearer way based on your suggestions and position it better.
>
> **Reply to “the formulation in Sec. 2.2”:** In this subsection, we want to say that we can also derive an informativeness measure for the non-probabilistic cases with the finite concept class **from the generalization bound in the PAC framework instead of from the generalization bound in the PAC-Bayesian framework as in Sec. 2.1**. Furthermore, for the three measures, PABI (definition 2.1), approximation of PABI (definition 2.2), and PABI in the PAC framework (equation (3)), we can see that **they are equivalent, i.e. $\hat{S}(\pi_0, \tilde{\pi}_0) = S(\pi_0, \tilde{\pi}_0) = S(\mathcal{C}, \tilde{\mathcal{C}})$**, in the non-probabilistic cases with the finite concept class, which indicating that **the approximation of PABI in definition 2.2 is reasonable**. We will modify this subsection to make it more clear based on your suggestions.
>
> **Reply to “the size of incidental supervision sets in the cross-domain NER”:** In our experiments, the main limitation is that we cannot use too many gold signals, or the performance of the lower bound will be too high, which makes incidental signals less meaningful. For simplicity, we let the number of incidental signals be nine times that of gold signals in all experiments, which makes the size of incidental signals also a little small. **We actually tried larger datasets for both gold and incidental signals (keeping the ratio between two sizes as 9) and the results are similar as long as the number of gold signals is not too large**. But you are right that we can increase the size of the incidental supervision signals and only keep gold signals small to see the value of incidental signals. **We will add this type of analysis in our later versions.**
>
> We will also incorporate your other suggestions accordingly.

---

> > ### Comment · AnonReviewer4 · 2020-11-16
> > **Sounds good, and raises a question**
> >
> > Thanks for your reply. Sounds good — I think increasing the ratio of incidental to gold supervision should makes sense here. But from looking through the explanation of PABI for transductive learning again, I have a couple more questions. Forgive me if I'm not fully understanding what's going on here.
> >
> > It seems that the formula for PABI in the transductive case is defined in terms of $\eta$, the noise rate of a perfect source-domain model on the target domain. This value itself is independent of the size of the source domain data. However, we estimate it making use of a trained model which is itself noisy. So our estimate of $\eta$ does depend on the amount of incidental supervision available, via the approximation. Even though the value of $\eta$ doesn't depend on this in principle.
> >
> > In practice, we expect the performance of a model in the target domain to increase with the amount of incidental supervision as well. And especially because of the assumption that $\eta_1 = \eta_1'$ made in the derivation (which also is a bit of a head-scratcher for me—isn't $\eta_1$ almost guaranteed to be lesser? Why make this assumption?), I would expect the proposed PABI measure to reflect this as well as the amount of incidental supervision increases, since higher performance in the source domain would yield a smaller estimate of the noise rate of $\hat{c}(\mathbf{x})$ on the target domain.
> >
> > Is this true? I guess a way to test it would be to vary the amount of source-domain supervision while keeping target-domain supervision constant, and test the correlation of the PABI estimate (computed using the approximation derived from each respective source domain set) with target-domain performance. If this correlation is there, while perhaps expected, it seems ... weird. Because it's happening through the error of the approximation in PABI, and not through PABI itself.
> >
> > From this I'm wondering if dataset size is a blind spot in certain instances of the PABI framework. Is the problem just needing "enough" source domain data to get a good estimate of the noise rate? Or does this mean it might have limitations more broadly around its ability to help us decide whether it'd be worth gathering/pretraining on more data? But, maybe it still works fine here just by virtue of the approximation? I'm not sure. Either way, it might a limitation worth mentioning (unless I missed it already in the paper). What do you think? Does this make sense?

---

> > > ### Author Response · Authors · 2020-11-16
> > > **Thanks for your good questions. You are correct that the size of incidental supervision signals should plan an important role in PABI.**
> > >
> > > **As printed at the beginning of Section 3, we actually discussed the impact of the size of incidental signals (also other important factors) in Appendix A.5.** There are three cases as follows (see more details in Appendix A.5):
> > > + **If $\tilde{m}$ is large enough, then PABI works.**
> > > + **If the sizes of all different incidental signals are the same, then PABI still works.** This is the case that our paper mainly focuses on. It means that we can still compare the benefits among different incidental signals with PABI because the relative improvement is independent of $\tilde{m}$ in this setting. It is worthwhile to note that PABI is mainly used to **compare the benefits for different signals not to predict the real generalization bounds**. Therefore, we care more about relative values instead of absolute values.
> > > + **If the incidental signals we are comparing are not large enough and have different sizes, we need to adjust PABI to incorporate that difference**. It seems that you are more interested in this case. This direction requires more explorations, which we postpone as our future work.
> > > **Sorry for our unclear presentation, we will present the impact of these factors in a clearer way in the final version.**
> > >
> > > **As for cross-domain signals**, we briefly answer your questions as follows:
> > > - **When $\eta$ is estimated, we can treat cross-domain signals as a special type of noisy data.** In this way, we can analyze the impact of the size of source data in three cases as mentioned above.
> > > - **The estimation of $\eta$:** You are correct that the estimation of $\eta$ depends on the size of incidental supervision signals. Under some assumptions, Eq. (4) serves as an unbiased estimator for $\eta$ (asymptotic analysis, i.e. with enough samples, the equation should hold), but **the concentration rate (or the variance) will depend on the size of source data**. It requires finer-grained analysis on the estimator in Eq. (4), which we postpone as our future work.
> > > - **The assumption $\eta_1 = \eta'_1$:** We add this assumption mainly because we want to estimate the $\eta$ only based on $\eta_1$ and $\eta_2$ which can be easily computed in practice.

---

> > > > ### Comment · AnonReviewer4 · 2020-11-16
> > > > **That answers my question! Sorry for missing it. But I'm still confused by Appendix A.5.**
> > > >
> > > > Great! Makes sense. I totally missed that comment at the beginning of the Section 3 when revisiting the paper just now.
> > > >
> > > > I admit I have a hard time following the development in Appendix A.5. I don't understand the justification for the formulas in the second and third case.
> > > >
> > > > In the second case, the equality $\frac{\ln | \tilde{\mathcal{C}}_\tilde{m} |}{\ln | \mathcal{C}_\tilde{m} |} = \frac{\ln | \tilde{\mathcal{C}} | }{\ln | \mathcal{C} |}$ is given for the case that the incidental supervision is constant between tasks. But shouldn't that be a statement about $\tilde{m}$ versus some $\tilde{m}'$ for two different sources of incidental supervision?
> > > >
> > > > In the third case, it looks especially weird to me because if $\frac{\ln|\tilde{\mathcal{C}}|}{\ln | \mathcal{C} |}$ is normally between 0 and 1, then I imagine multiplying by $\tilde{m}$ would result in a negative value under the square root for sufficiently large $\tilde{m}$. In any case shouldn't it reduce to the formula given in case 1 when $\tilde{m} \to \infty$?
> > > >
> > > > I'll totally allow that this is just my confusion here because, as I said, I'm not familiar with PAC-Bayesian theory, and I might not be correctly following all of your derivations.

---

> > > > > ### Author Response · Authors · 2020-11-16
> > > > > **Thanks for your questions! We will update a new version soon to better present our paper.**
> > > > >
> > > > > Actually, there are several typos in the submitted version of the Appendix. Sorry about that, and **we will update a new version soon by fixing these typos and also some suggestions from you and other reviewers**. We briefly mentioned the new version of Appendix A.5 and hope it will help you have a better understanding of the impact of the size of incidental signals.
> > > > >
> > > > > A more realistic PABI is based on $\tilde{\mathcal{C}}$ with $\tilde{m}$ examples as
> > > > > $S(\mathcal{C}, \tilde{\mathcal{C}}) = \sqrt{ \frac{\ln |\mathcal{C}\_{\tilde{m}}| - \ln{|\tilde{\mathcal{C}}\_{\tilde{m}}|}}{\ln{|\mathcal{C}|}}} = \sqrt{\frac{\ln |\mathcal{C}\_{\tilde{m}}| - \ln |\tilde{\mathcal{C}}\_{\tilde{m}}|}{\ln{|\mathcal{C}\_{\tilde{m}}|}} \times \frac{\ln{|\mathcal{C}\_{\tilde{m}}|}}{\ln{|\mathcal{C}|}}} = \sqrt{(1 - \frac{\ln |\tilde{\mathcal{C}}\_{\tilde{m}}|}{\ln{|\mathcal{C}\_{\tilde{m}}|}}) \times \frac{\ln{|\mathcal{C}\_{\tilde{m}}|}}{\ln{|\mathcal{C}|}}} = \sqrt{(1 - \frac{\ln |\tilde{\mathcal{C}}|}{\ln{|\mathcal{C}|}}) \times \frac{\ln{|\mathcal{C}\_{\tilde{m}}|}}{\ln{|\mathcal{C}|}}}.$
> > > > >
> > > > > Note that the ratio of the intrinsic information in incidental signals is independent of the size $\tilde{m}$, so $\frac{\ln | \tilde{\mathcal{C}}\_{\tilde{m}}|}{\ln{|\mathcal{C}\_{\tilde{m}}|}} = \frac{\ln | \tilde{\mathcal{C}}|}{\ln{|\mathcal{C}|}}$ holds for incidental signals. For example, $\frac{\ln | \tilde{\mathcal{C}}\_{\tilde{m}}|}{\ln{|\mathcal{C}\_{\tilde{m}}|}} = \eta_p$ for partial data with partial rate $\eta_p$, which doesn't depend on the size $\tilde{m}$.
> > > > >
> > > > > (1) When $\tilde{m}$ is large enough, $S(\mathcal{C}, \tilde{\mathcal{C}}) = \sqrt{1 -  \frac{\ln{|\tilde{\mathcal{C}}|}}{\ln{|\mathcal{C}|}}}$.
> > > > >
> > > > > (2) When the sizes of different incidental signals are all $\tilde{m}$, the informativeness measure $S(\mathcal{C}, \tilde{\mathcal{C}}) = \sqrt{(1 - \frac{\ln |\tilde{\mathcal{C}}|}{\ln{|\mathcal{C}|}}) \times \frac{\ln{|\mathcal{C}\_{\tilde{m}}|}}{\ln{|\mathcal{C}|}}}$ is independent of $\tilde{m}$ **because $\frac{\ln{|\mathcal{C}\_{\tilde{m}}|}}{\ln{|\mathcal{C}|}}$ is the same constant for different incidental signals**. **Note that $(1 - \frac{\ln |\tilde{\mathcal{C}}|}{\ln{|\mathcal{C}|}})$ will be different for different signals.**
> > > > >
> > > > > (3) The incidental signals we are comparing are not large enough and have different sizes, we need to use $S(\mathcal{C}, \tilde{\mathcal{C}}) = \sqrt{(1 - \frac{\ln{|\tilde{\mathcal{C}}|}}{\ln{|\mathcal{C}|}}) \times \frac{\tilde{m}}{|\mathcal{V}|^n}}$ to incorporate that difference. We can replace $|\mathcal{V}|^n$ with some reasonable $M$, e.g. the largest size of incidental signals, to make PABI in a larger range of values in $[0, 1]$.  This direction requires more explorations, which we postpone as our future work.

---

> > > > > > ### Comment · AnonReviewer4 · 2020-11-19
> > > > > > **Thanks!**
> > > > > >
> > > > > > I think I'm tracking. Makes sense. Thanks so much for your detailed replies!

---

> > > > > > > ### Author Response · Authors · 2020-11-23
> > > > > > > **Thanks!**
> > > > > > >
> > > > > > > Thanks a lot for your suggestions! We really appreciate your efforts in reviewing our paper and we believe that we together can achieve a better paper with your insightful comments. **We modified our paper in the revision,  based on your suggestions except for the scope suggestion. Please let us know if you think the revision is not enough. As for the scope suggestion, it is an excellent suggestion and we need more time to reposition our paper with your suggestion. The scope issue will be handled in our final version.**

---

### Official Review · AnonReviewer3 · 2020-10-28
**Unified measure for alternative supervision signals**

**Rating:** 7
**Confidence:** 3

**Review:**


#### Summary

This paper proposes a unified measure for the informativeness of incidental signals (ie, not standard ground truth supervised labels) derived from the PAC-Bayesian theoretical framework. Instantiations of the score are derived for a variety of these signals, and experiments show good agreement between the measure and true performance improvements.

#### Strong and weak points

This problem setting is well-positioned as complementary to the growth in "alternative supervision" in both research and industry. Besides the directions identified in the paper, one could easily imagine using this kind of a measure in ML applications as a tool to help guide economic decisions about what kinds of datasets or annotations to pursue. The explanatory potential of this approach with respect to observed gains using incidental signals is exciting as well, especially the agreement with empirical findings from Mayhew 2019.

I found Figure 1 to give helpful context, and I found the core technical content in Section 2 to be clear and precise.

The experimental results were a bit intricate to follow. A key result is Figure 2f and the associated correlations, which show strong correlation between the PABI scores and true performance improvements, this could perhaps be higlighted or emphasized more. Likewise the meaning of Figure 3 is a bit obscured by the poor correlations of the
baselines.

One weakness of the evaluation was that, while the Related Work coverage seemed sufficient, only Gururangan 2020 is included in the experiments. Of course the other approaches have the limitations well-captured in Table 1, but it would have been nice to have some restricted experiments crafted in order to give direct comparisons.

The supplemental appendix was comprehensive with respect to theoretical derivations and experimental details.

#### Recommendation (accept or reject) with one or two key reasons for this choice.

I would recommend to accept, the work represents an advance across both theory and practice on an important problem.

#### Supporting arguments

The work leverages a well-studied framework to answer important questions about understanding the utility of non-standard supervision signals, enabling us to reason in a unified way about varied kinds of these signals as well as their combinations. Experimental results


#### Questions to clarify / additional evidence required

The approximation in Definition 2.2 was a little strange for me, and seemed kind of circular: for calculating our approximate PABI, we are approximating the target (gold) distribution with our approximatively improved prior ($\tilde{pi_0}$)? Is there anything we can say about how good/accurate this approximation is?

Section 3.2: "much cheaper" - how or why would we say this is true, can we quantify it? Or are there cites to see?

Is it possible to frame the PABI measures in terms of testable hypotheses about true generalization error, or are the bounds too loose in practice to say anything meaningful here?

#### Additional feedback to improve

Space permitting, a small diagram of the mappings between different domains and the restrction trick would make Section 3.2 much clearer. Another possibility is some symbol table to keep straight which versions of $c()$ correspond to gold vs silver, incidental, etc.

The code was great to see as well but is missing dependencies:

- seqeval
- tqdm
- transformers

I might suggest adding a requirements.txt or similar.

---

> ### Author Response · Authors · 2020-11-13
> **Thanks for your insightful comments! In particular, we really appreciate it that you took a look at our code and gave us several suggestions on it.**
>
> **Reply to “the weakness of the evaluation”:** **We actually already compared PABI with most related work in Table 1 in our experiments.** The experimental comparisons are shown in Figure 2 and Figure 3. Specifically, Figure 2(a) compares the PABI with the partial rate used in CST’11, HH’12, and LD’14 (shown in Table 1); Figure 2(b) compares the PABI with the noisy rate used in AL’88, NDRT’13,  and RVBS’17 (shown in Table 1); Figure 2(c) compares the PABI with the mutual information used in B’17 (shown in Table 1); Figure 3 compares the PABI with the vocabulary overlap used in GMSLBDS’20 (shown in Table 1). **Sorry for our unclear presentation, we will present the comparisons in a clearer way in the final version.**
>
> **Reply to “the approximation in Definition 2.2”:** It’s hard to directly quantify the quality of this approximation. However, for the three measures, PABI (definition 2.1), approximation of PABI (definition 2.2), and PABI in the PAC framework (equation (3)), we can see that they are equivalent, i.e. $\hat{S}(\pi_0, \tilde{\pi}_0) = S(\pi_0, \tilde{\pi}_0) = S(\mathcal{C}, \tilde{\mathcal{C}})$, in the non-probabilistic cases with the finite concept class. We believe that **the equivalence among three measures in the non-probabilistic cases with the finite concept class**, such as partial labels in Section 3.1, can serve as good evidence that this approximation is reasonable. Furthermore, **the effectiveness of this approximation in NLP applications** also indicates the quality of this approximation.
>
> **Reply to “much cheaper in Sec. 3.2”:** For example, given n source domains and n target domains, our goal is to select the best source domain for each target domain. **If we use the joint training, we need to train $n \times n= n^2$ models.  But with PABI, we only need to train $n+n=2n$ models.** Furthermore, for each model, joint training on the combination of two domains **requires more time** than single-domain training used in PABI. In this situation, we can see that PABI is much cheaper than building combined models with joint training.
>
> **Reply to “PABI and the generalization bound”:** Great suggestions and it is actually our future work. It might be too hard to directly link PABI with the generalization bound for all types of incidental signals, but **it is possible to link it to the generalization bound for some specific types**. For example, for partial and noisy labels, PABI can directly be expressed in the generalization bound as in CST’11, HH’12, LD’14, AL’88, NDRT’13, RVBS’17, VW’17, WNR’20 (shown in Table 1). The main difficulties are in constraints and auxiliary signals, and we postpone it as our future work.
>
> **Reply to “the dependencies in the code”:** Thanks for checking our code. We will add a requirement.txt with those missing dependencies accordingly.
>
> **We modified our paper in the revision, based on your suggestions. Please let us know if you think the revision is not enough.**

---

### Official Review · AnonReviewer2 · 2020-10-30
**The paper explores a new and exciting territory, but needs a deeper analysis to support the connection with the PAC-Bayes theory.**

**Rating:** 5
**Confidence:** 3

**Review:**

The use of PAC-Bayes theory for NLP tasks is rare. Although I know little on NLP, the paper proposition to leverage on  PAC-Bayes for evaluating the benefit of various incidental supervision signals seems promising. However, even if the empirical results are good, the connection between PAC-Bayes and the proposed informativeness measure (named PABI) is vague. The paper needs to better situate the proposed analysis compared to classical PAC-Bayesian generalization risk bounds.

**Section 2 contains many assertions that are questionable.**
1. *"The training samples [are] generated i.i.d."*: It is the case for most PAC-Bayes analysis, but I wonder to which extent this assumption holds for the NLP problems studied as experiments. In a sentence, words are highly dependent on each other.
2. *"In the common supervised learning setting, we usually assume the concept that generates data comes from the concept class"*: This is a surprising claim as the **PAC**-Bayes framework differs from the Bayesian one namely by the fact that we usually don't need to make assumptions about the data-generating distribution other than being i.i.d. In particular, the model does not need to be well specified. This makes me wonder if PABI would not better fit in the purely Bayesian framework (see other comments below).
3. *"the generalization bounds in both PAC-Bayesian and PAC frameworks have the square root function"* : There exist several forms of the PAC-Bayes theorem in the literature, not only the square root ones (e.g., Seeger 2002). In fact, the square root bounds are not the tightest, particularly when the model is well specified.
**Is PABI really backed by PAC-Bayes theory?**
As far as I understand, the procedure PABI is only remotely inspired by the PAC-Bayes bound,  but is not truly justified by it. No PAC-Bayes bounds are fully optimized; PABI borrows from PAC-Bayes the sole idea of relying on the KL between distribution. For this reason, I think that the introduction sentence "Previous attempts are either not practical or too heuristic" is harsh, because the proposed method turns out to be a heuristic too.

**Is PABI a more Bayesian method than a PAC-Bayes one?**
I wonder if one could not do the same analysis in a fully Bayesian setting, maximizing a Bayesian information criterion. This should be appropriate since PABI and the Bayesian setting assume that the model is well specified. Note that there is a direct link between the Bayesian Marginal Likelihood and the PAC-Bayes generalization bound (e.g., Germain, et al., 2016: "PAC-Bayesian Theory Meets Bayesian Inference.")

Overall, I think that the paper explores a new and exciting territory, but needs a deeper analysis to support the connection with the PAC-Bayes theory.

---

> ### Author Response · Authors · 2020-11-13
> **Thanks for your insightful comments!**
>
> **Reply to “Is PABI really backed by PAC-Bayes theory”:** We agree that the connection between PAC-Bayes and PABI is not strong, and we will clarify this point later; we will especially change our wording from “PAC-Bayesian based”, which might be misleading to some readers, to “PAC-Bayesian motivated”. However, we want to highlight that **whether PABI is “based on” or “motivated by” PAC-Bayes theory plays a minor role in our contribution**.
>
> First, as also noted by other reviewers, in the field of NLP, **existing measures of “usefulness of an indirect signal” have even weaker connections to machine learning theory, and researchers have been having a hard time finding theoretical groundings**. E.g., Gururangan et al. (2020) used vocabulary overlap to estimate similarities of different domains of text; Ning et al (2019) used mutual information to estimate the benefits of partial annotations. Compared to prior works in NLP, PABI is already *better* grounded in theory.
>
> Second, even without a fully derived generalization bound showing the theoretical soundness of PABI, **PABI is still shown to be an effective unified measure as an estimate of the benefit of imperfect data as shown in our experiments**. This empirical value is also important as pointed out by other reviewers.
>
> Third, it’s important to notice that **most advanced theories cannot be easily used in practice**. For example, Baxter (2000) provides a good theory for multi-task learning, but it's still unclear how to use this theory in practice. Specifically, it is difficult to model the data structure of natural language text and capture the properties of deep neural networks, which prevents the use of most well-developed theories. More examples can be found in the related work.
>
> **In summary, we agree the connection between PAC-Bayes and PABI is not fully developed, but the real contribution is that “we find a well-motivated and unified measure that is shown to be effective in various experiments.”**
>
> **Reply to “the training samples [are] generated i.i.d”:** You are right. The i.i.d. assumption doesn’t hold for most NLP problems, and a lot of efforts have been made to capture these dependencies, as in CRF [1] and ILP [2]. In our framework, **we treat the dependency among words as a special type of incidental supervision signals, i.e. constraints**, and want to measure the benefit of this dependency. In other words, the baseline in our setting will be the data without any extra information (including the dependency among words), which is common in NLP applications. We want to highlight that this type of dependency is crucial for structured prediction but its benefit is not well understood, and our framework might be able to shed some light on it.
>
> **Reply to “the square root function”: The square root is not crucial for our framework**, because our goal is to compare the benefits among different incidental supervision signals, where **the relative values are expressive enough**.  In this sense, any strictly increasing function in [0, 1] over the current formulation would be acceptable. We use the square root mainly because the Pearson correlation looks better with the square root in our experiments.
>
> **Reply to “Is PABI a more Bayesian method than a PAC-Bayes one”:** Thanks for pointing out this (maximizing a Bayesian information criterion) perspective and the connection between Bayesian Marginal Likelihood and PAC-Bayesian generalization bound. We think it is a good question to explore but we need to carefully review the reference paper and relevant literature in order to make a precise clarification. Generally speaking, PABI measures how much the incidental signals can improve the prior distribution with respect to the gold posterior. **The reason why we chose the PAC-Bayes framework is that it allows us to link PABI to the performance measure**. It is possible to derive PABI in a fully Bayesian setting (can be an interesting future work). We do not think this will affect the soundness and empirical value of PABI: to provide a unified informativeness measure that can handle various types of incidental signals, motivated by theory and at the same time, easily computed in real-world applications.
>
> **We modified our paper in the revision, based on your suggestions. Please let us know if you think the revision is not enough.**
>
> [1] John Lafferty, Andrew McCallum, and Fernando C.N. Pereira. Conditional Random Fields: Probabilistic Models for Segmenting and Labeling Sequence Data. Proceedings of the Eighteenth International Conference on Machine Learning, 2001.
>
> [2] Dan Roth and Wen-tau Yih. A linear programming formulation for global inference in natural language tasks. ILLINOIS UNIV AT URBANA-CHAMPAIGN DEPT OF COMPUTER SCIENCE, 2004.

---

### Author Response · Authors · 2020-11-23
**Thanks for your suggestions!**

**We modified our paper in the revision based on the suggestions from reviewers. The revised parts are in blue color. Please let us know if you think the revision is not enough.**

---

### Decision · Program_Chairs · 2021-01-07
**Final Decision**

**Decision:**

Reject

**Comment:**

This paper first makes the observation that incidental supervisory data can be used to define a new prior from which to calculate a PAC-Bayes generalization guarantee.  This observation can be applied to any setting where there is unsupervised or semi-supervised pre-training followed by fine-tuning on labeled data.  The PAC-Bayes bound is valid when applied to the fine-tuning.  For example, one could use an L2 bound (derived from PAC-Bayes) on the difference between the fine-tuned parameters and pre-trained parameters.

But the paper proposes evaluating the value of pre-training before looking at any labeled data. Let $\pi_0$ be the prior before unsupervised or semi-supervised training and let $\tilde{\pi}$ be the prior after pre-training.  The paper proposes using the entropy ratio $H(\pi_0)/H(\tilde{\pi})$ as a measure of the value of the pre-training.  As the reviewers note, this is not really related to PAC-Bayes bounds.  Furthermore, it is clearly possible that the pre-training greatly focuses the prior but in a way that is detrimental to learning the task at hand.

I have to side with the reviewers that feel that this is below threshold.